# CROSS-LINGUAL MULTIMODAL RETRIEVAL-AUGMENTED GENERATION FOR OPEN QUESTION ANSWERING IN TAMIL AND YORUBA

## ABSTRACT

As large language models (LLMs) with retrieval-augmented generation (RAG) gain traction in multimodal knowledge-base question answering (KBQA), concerns about their transfer to low-resource languages (LRLs) remain unaddressed. We introduce **LR-MMQA**[1], a benchmark assessing multimodal cross-lingual retrieval and reasoning under the challenges of LRLs. Using a state-of-the-art LLM, we translated the hardest questions from WebQA and MultimodalQA, creating a dataset that stresses cross-evidence aggregation and multi-hop inference. We also introduce **XM-RAG**, a cross-lingual multimodal RAG pipeline optimized for LRLs, which achieves 38.1 answer accuracy overall, over 6.3 points higher than the next best baseline. Our findings expose significant biases and discrepancies in existing systems, with LR-MMQA highlighting specific failure points. Notably, XM-RAG's performance on LR-MMQA is far below top models on English datasets (WebQA: 64.4, MultimodalQA: 73.48 answer accuracy), demonstrating that **current methods still fail at complex, real-world tasks in LRLs**. By releasing LR-MMQA and XM-RAG, we provide a resource to evaluate and address these gaps and guide progress toward equitable multimodal KBQA.

## 1 INTRODUCTION

In recent years, Large Language Models (LLMs) have made significant strides in Knowledge Base Question Answering (KBQA) through Retrieval Augmented Generation (RAG) (Lewis et al., 2020; Xu et al., 2024; Luo et al., 2024), a paradigm that increasingly leverages multimodal retrieval from vast corpora to demonstrate improved accuracy over text-only methods (Suri et al., 2025; Chen et al., 2022; Ling et al., 2025; Yan & Xie, 2024). Despite these achievements, retrieval models still struggle to answer knowledge-based questions accurately, fluently, and completely in low-resource languages (LRL) due to limited training data and a lack of high-quality retrieval content in these languages (Qi et al., 2025; Rogoz & Lupaşcu, 2025). Various methods, such as translate-then-retrieve, have been developed to address this problem (Asai et al., 2021) and have been further enhanced by using a multilingual encoder to embed the query in a multilingual semantic space to be used for retrieval from a high-resource language (HRL) corpora (Asai et al., 2022). This shifts reliance from the inadequate knowledge present in LRLs to the more comprehensive knowledge in HRLs. This method has been recently augmented via the addition of an image encoder and multimodal retrieval framework, expanding the scope of questions that can be answered correctly (Li & Ke, 2025).

However, this solution for multimodal KBQA in LRLs is an extremely basic RAG pipeline that underperforms compared to the state-of-the-art seen in high-resource language (HRL) systems (Mei et al., 2025). State-of-the-art unimodal frameworks for LRLs exist, but they lack the crucial multimodal processing needed to accurately reflect human communication and information understanding (Baltruaitis et al., 2019). While HRLs have advanced significantly in multimodal KBQA, LRL progress is years behind (Rogoz & Lupaşcu, 2025). Furthermore, evaluation of multimodal retrieval for open KBQA in LRLs is impossible, as there are no datasets with LRL questions that require multimodal understanding and retrieval for their answers (Rogoz & Lupaşcu, 2025). This lack of a

---

[1]You can find the dataset here: https://huggingface.co/datasets/anonymous132145/LR-MMQA

benchmark makes it impossible to identify and address the shortcomings in current models, thereby perpetuating the performance gap between languages and preventing significant progress.

To enable comprehensive evaluation and critical advances in this area, we introduce LR-MMQA, the first multimodal, cross-lingual open KBQA benchmark for LRLs, featuring 718 questions in Yoruba and Tamil, with ground-truth documents in english. This dataset is designed to require multimodal query understanding as well as multimodal and cross-lingual retrieval for complete answers, with all translations validated by native speakers to ensure accuracy.

We also propose XM-RAG, a novel multimodal RAG baseline. XM-RAG is designed to enable accurate and grounded KBQA for LRLs by directly encoding LRL queries and employing a cross-lingual, multimodal retrieval mechanism from a high-resource knowledge base. The retrieved evidence is reranked using a state-of-the-art learned reranker and then summarized and fused via a refinement and fusion layer. The fused multimodal evidence is then used to generate high-quality answers in the user's original language.

Overall, XM-RAG significantly outperforms existing baselines on LR-MMQA in both accuracy and F1 without fine-tuning. By introducing this framework and benchmark, we aim to enable complete, accurate, and accessible open KBQA across languages in a lightweight and modular fashion. **Our main contributions are:**

- **LR-MMQA**, the first LRL KBQA benchmark requiring multimodal query understanding and multilingual multimodal retrieval from Tamil and Yoruba. LR-MMQA enables a finer analysis of RAG models in low-resource settings, revealing significant weaknesses in cross-lingual retrieval, multi-hop reasoning, and answer synthesis, and guiding progress toward equitable QA.
- **XM-RAG**, a multimodal RAG baseline designed for accurate, fluent, and grounded Knowledge Base Question Answering in low-resource languages, leveraging cross-lingual multimodal retrieval from high-resource multimodal knowledge bases.
- We show that XM-RAG significantly **improves performance in terms of both accuracy and retrieval** for KBQA in both LRLs.

## 2 RELATED WORKS

**Multimodal Retrieval-Augmented Generation for Knowledge Base Question Answering** Despite the many advances of LLMs with multimodal RAG, they still struggle to use external knowledge and unseen data, both of which are necessary for KBQA (Zhang et al., 2024). RAG addresses this issue by retrieving external evidence from a corpus of knowledge, in turn increasing accuracy and grounding of model responses to knowledge-base questions (Lewis et al., 2020). These results have seen further improvements due to multimodal retrieval. Methods such as Multimodal Multihop, a methodology used to gather data from multiple sources to formulate an answer, show evidence of these promising results when built upon baseline models (Yarabelly, 2025). Multimodal retrieval allows for the vast multimodal evidence to be leveraged to answer questions that cannot be fully answered with only text. MuRAG (Chen et al., 2022) does this by treating images as visual tokens. RA-BLIP (Ding et al., 2024) projects retrieved text and images into a shared space before fusion. No matter how these methods treat images, they only retrieve content from the language of the query, meaning the quality of the answers is dependent on the quantity and quality of retrievable evidence present, both of which are lacking in low-resource languages.

**Cross-Lingual Retrieval** Work has shown that retrieval models struggle in LRLs due to lack of high-quality retrieval content (Qi et al., 2025; Rogoz & Lupaşcu, 2025). A commonly explored solution to this problem has been a pipeline in which the original LRL query is used to retrieve content from a high-resource knowledge base (KB). That content is used to generate an answer in the original LRL. Quite a lot of work has been done exploring this solution. The most common approach to this is the translate-then-retrieve pipeline, as seen in XOR-RETRIEVE (Asai et al., 2021). This approach translates the query to a high resource language, uses this embedding to retrieve text evidence, and feeds it into a multilingual pre-trained model to generate an answer in the LRL. A substantial improvement on this approach can be seen in CORA, where a multilingual embedding of the query is used, reducing errors that arise with machine translation (Asai et al., 2022). This direction, leveraging multilingual models like BERT for direct cross-lingual information

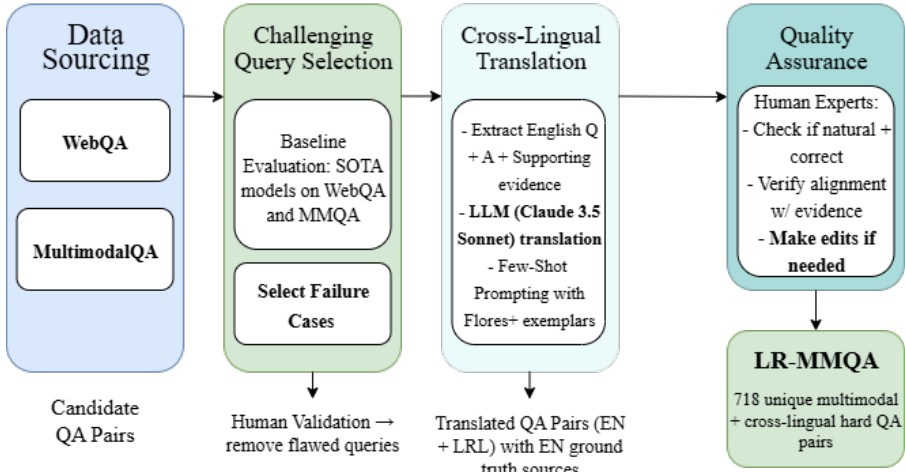

Figure 1: Flowchart of LR-MMQA creation from WebQA and MultimodalQA using Claude 3.5 Sonnet.

retrieval, has been explored in various works, demonstrating effectiveness in matching queries across languages (Jiang et al., 2020). The embedded query is then fed into a pre-trained retrieval algorithm and the retrieved evidence is fed into a multilingual auto-regressive generation model to produce an answer. However, all these works only take in, reason on, and retrieve text, significantly limiting the type of questions they can answer accurately.

**Multimodal Reasoning and Retrieval Knowledge Base Question Answering Benchmarks** Many KBQA benchmarks contain or are entirely composed of questions that require retrieval and understanding across data modalities (Chang et al., 2022; Talmor et al., 2021; Marino et al., 2019). However, these datasets solely contain questions and answers in high-resource languages (HRLs). On the other hand, there are KBQA benchmarks that contain questions and answers for LRLs, but only provide the dataset in a single modality, normally text (Sawczyn et al., 2024; Rohera et al., 2024; Longpre et al., 2020). While VQA evaluation in low-resource languages has been studied (Nguyen et al., 2023; Kim et al., 2024; Salazar et al., 2025), evaluation of multimodal reasoning with knowledge intensive question answering (like the multi-hop retrieval required by LR-MMQA) in these languages remains largely unexplored, which is the focus of LR-MMQA.

## 3 DATASET

### 3.1 DATASET OVERVIEW

**LR-MMQA** is a benchmark designed to rigorously evaluate RAG systems on multimodal and cross-lingual understanding and retrieval for low-resource open KBQA. Curated from two high-resource datasets, it comprises 718 unique multimodal questions specifically selected for their difficulty, representing cases where state-of-the-art models currently fail. Questions were first translated from Standard American English (SAE) into Tamil and Yoruba using an LLM, then post-edited by native-speaker volunteers to ensure fluency and correctness. See Figure 1 for a visual overview of LR-MMQA creation. The questions require multilingual reasoning and retrieval, as the ground-truth documents are not in the query's language. This design simulates a real-world QA environment where comprehensive sources are often unavailable in low-resource languages.

### 3.2 DATA COLLECTION AND PREPARATION

**Data Sourcing** Our dataset is derived from the WebQA dataset (Chang et al., 2022) and a subset of the MultimodalQA dataset (Talmor et al., 2021), both of which are standardized collections of open-domain knowledge-seeking queries. Specifically, these datasets contain queries that require models to retrieve and reason over images, text, or both. Using WebQA and MultimodalQA ensured our

analysis was based on authentic, knowledge-seeking questions, enabling question-answering that feels natural and relevant, even when extended to low-resource settings.

**Challenging Query Selection** To achieve rigorous evaluation, we pre-filtered by running a few existing baselines on WebQA and MultimodalQA. We ran SKURG (Yang et al., 2023) and RAMQA (Bai et al., 2025) on both datasets, selecting these two publicly available multimodal retrieval frameworks as they represent the highest performing models on the respective datasets. We run these models to establish a higher performance ceiling and identify questions that remain challenging even for state-of-the-art systems, thereby defining a more robust set of "hard" examples. We then selected questions based on their "failure" status between the two models, where a failure represents an average WebQA QA score (composite metric of question accuracy and BARTScore (Yuan et al., 2021)) or MultimodalQA accuracy score of 0. Humans then validated each of these questions to ensure that they contained no errors that made them unanswerable, such as the answer no longer relating to the question. All flawless questions were selected for translation. This comprehensive inclusion directly captures every instance where state-of-the-art systems demonstrably fail, aligning precisely with the benchmark's objective. Table 1 shows a specific breakdown of the selected queries.

| Origin | Text | Image | Image + Text |
|---|---|---|---|
| WebQA | 106 | 490 | 0 |
| MultimodalQA | 41 | 67 | 14 |
| **Total** | **147** | **557** | **14** |

Table 1: Origin and required retrieval modalities for the selected question-answer pairs. (**Total unique questions: 718**)

### 3.3 Cross-Lingual Translation and Annotation

**Language Selection** The question-answer pairs in LR-MMQA are in the two low-resource languages (LRLs) of Tamil and Yoruba. Other than being the languages that the team members speak fluently, Tamil and Yoruba are highly semantically diverse, enabling a broad and representative evaluation of cross-lingual capabilities and understanding.

**Translation Protocol** For each selected sample in the English benchmarks, we extract the English Question and English gold answer, along with any supporting materials (i.e., if the dataset includes questions with images). To translate QA pairs from SAE to each of the two LRLs, we employed a few-shot prompting strategy (Brown et al., 2020) informed by examples from FLORES+, a high-quality dataset containing parallel examples of human translated sentences across languages, including Tamil and Yoruba. Prior work has shown that exemplar-based prompting improves multilingual translation quality in LLMs (Lin et al., 2022). We used three exemplar translations from FLORES+ per language. Utilizing Claude 3.5 Sonnet, the LLM was prompted to rewrite the QA pairs into Tamil and Yoruba, informed by the examples. This approach ensures that translations maintain linguistic authenticity and respect the semantic nuances of the target language. Detailed examples of these prompts can be found in Appendix A. Claude 3.5 Sonnet is used because previous work has shown that it has remarkable resource efficiency and outperforms state of the art neural machine translation (NMT) on translation tasks (Enis & Hopkins, 2024). Furthermore, past work has also shown that LLM translation is superior to NMT in terms of how closely it resembles human translation (Sizov et al., 2024), an important quality as questions and answers must appear genuine to truly gauge a model's ability to answer questions in the LRL. Samples of translated queries can be found in the Appendix B.

**Structured Entries** After translation, the data points for the benchmark were prepared. Every data point in LR-MMQA can be expressed as the tuple $(Q_{EN}, Q_{LRL}, E_{MM}, A_{EN}, A_{LRL}, S_{GD})$, where $Q_{LRL}$ is the low-resource language question with its parallel English translation $Q_{EN}$, $E_{MM}$ represents the supporting evidence (images), $A_{LRL}$ is the low-resource language gold answer with its parallel English translation $A_{EN}$, and $S_{GD}$ are the ground truth documents needed to accurately answer the questions. There is one data point for each of the two languages, meaning LR-MMQA is comprised of 1436 of these data points. A sample data point can be found in Appendix C.

### 3.4 QUALITY ASSURANCE AND VALIDATION

To ensure no errors persist in the dataset, human bilingual experts verified that the LRL QA pairs accurately relate to the supporting evidence, as translation can subtly alter word meanings, potentially rendering a question unanswerable by the provided context or misaligning it with the gold answer. If they noticed any errors in translation, they made necessary corrections.

Additionally, we conducted systematic translation quality evaluation on a representative sample (F). Two native speakers independently assessed 150 translations per language using 10-point Likert scales for adequacy and fluency, achieving substantial inter-annotator agreement ( = 0.76 overall) and high quality scores (8.2 adequacy, 8.0 fluency), confirming the effectiveness of our LLM-assisted translation.

## 4 XM-RAG

### 4.1 OVERVIEW

XM-RAG is a cross-lingual, cross-modal RAG pipeline for knowledge-based question answering (KBQA) in LRLs. It retrieves text and image evidence from high-resource corpora and generates answers directly in the users language. The system follows a modular design suitable for low-resource settings, relying on off-the-shelf encoders and the strong multilingual multimodal retrieval capabilities of M-CLIP.

### 4.2 INPUT PROCESSING

Given an LRL question $q$ (optionally with an image), we perform lightweight language identification (FastText) to attach a language tag (e.g., <am>, <yo>, <ta>). While not required for LR-MMQA evaluation, this enables broader applicability. The question is encoded without translation using M-CLIP to obtain a unit-normalized text embedding $\mathbf{q}_{\text{text}}$, and input images are encoded into $\mathbf{q}_{\text{img}}$ using the same backbone:

$$\mathbf{q}_{\text{text}} = \text{M-CLIP}_{\text{text}}(q), \quad \mathbf{q}_{\text{img}} = \text{M-CLIP}_{\text{vision}}(I_{\text{in}}).$$

### 4.3 CROSS-MODAL RETRIEVAL

We maintain separate FAISS indices for HRL text corpora (benchmark texts, Wikipedia, web snippets) and images. Using $\mathbf{q}_{\text{text}}$ and $\mathbf{q}_{\text{img}}$, we perform $k$-NN search to obtain top-$K$ text candidates $\{(d_i, s_i^{\text{text}})\}$ and top-$K$ image candidates $\{(v_j, s_j^{\text{img}})\}$.

Here, $\mathbf{d}_i$ denotes the $i$-th retrieved text passage and $\mathbf{v}_j$ the $j$-th retrieved image. Their similarity scores to the query in the shared embedding space are written as $s_i^{\text{text}}$ and $s_j^{\text{img}}$, respectively.

IVF/Flat indexing provides scalable retrieval. For LRL-MM-QA evaluation, HRL ground-truth documents come from MultimodalQA and WebQA.

### 4.4 CROSS-MODAL RERANKING

We compute reranking scores combining textual and visual evidence:

$$S(d) = \alpha \cdot s^{\text{text}}(d) + \beta \cdot \max_{v \in \mathcal{N}(d)} s^{\text{img}}(v) + \gamma \cdot \phi(d), \tag{1}$$

where $\mathcal{N}(d)$ are images co-occurring with passage $d$, and $\phi(\cdot)$ is a lightweight heuristic feature (e.g., language or answer-type cues). Parameters are fixed for the baseline; an MLP reranker is optional.

### 4.5 MULTIMODAL FUSION AND EVIDENCE COMPRESSION

We apply late fusion by selecting top-$n$ passages and top-$m$ images and compressing them into a concise context:

- BLIP-2 produces bilingual (LRL-tagged) two-sentence visual summaries.

- HRL passages are truncated using rare-caption filtering and deduplication.

This yields compact, salient evidence with low computational overhead.

### 4.6 MULTILINGUAL ANSWER GENERATION

The fused context and original LRL question are provided to an mT5 generator with the language tag prepended:

$$\text{Answer} = \text{mT5}([q_{\text{LRL}}] \oplus \text{evidence}_{\text{fused}}).$$

The tag enforces output in the target LRL and prevents accidental HRL translation. The model attends to both textual evidence and BLIP-2 visual summaries.

### 4.7 TRAINING AND INFERENCE CONFIGURATION

XM-RAG runs zero-shot with off-the-shelf components. All embeddings are unit-normalized, and FAISS IVF indices use an $n_{\text{probe}}$ value tuned to each corpus size. We apply strict token caps to control latency and memory usage, and the model operates without any task-specific fine-tuning.

## 5 RESULTS AND ANALYSIS

### 5.1 METRICS

To assess model performance on LR-MMQA, we used a combination of quantitative and qualitative metrics:

- Retrieval Metrics: Precision, recall, and F1 scores were used to evaluate the quality of retrieved documents
- Accuracy: Token overlap scores were used to measure the accuracy and coherence of generated outputs.

LR-MMQA utilizes a Full-Wiki evaluation, meaning the model must retrieve the correct answer from the entire Wikipedia corpus. This differs from datasets like WebQA and MultimodalQA that provide a small set of candidate documents, thus eliminating the need for a separate retrieval step. In the LR-MMQA setting, there are no distractors to contend with, as the primary task is to find and extract the correct information from a full knowledge base in a different language than the query, simulating a real-world scenario.

### 5.2 BASELINES

Given the absence of a publicly available true multimodal, multilingual baseline model that directly addresses all aspects of our research, we selected a set of representative baselines. These models were chosen to provide a comprehensive comparison against our proposed approach by evaluating its performance across different modalities and languages. This multifaceted evaluation serves to expose the specific gaps in current models that our approach aims to address.

**Text-Only Cross-Lingual RAG Baseline** This baseline represents the current state-of-the-art in Text-Only RAG, but also serves to highlight its core limitation: the inability to process non-textual information. Although it demonstrates advanced cross-lingual generation by retrieving from a high-resource language and answering in a target language, its text-only nature makes it fundamentally inadequate for the real-life questions posed in the LR-MMQA benchmark, which requires a model to reason across different modalities. Its inclusion demonstrates that even sophisticated Text-Only models fail when faced with the real-world complexity of multimodal tasks.

**Multimodal Monolingual RAG Baseline** This baseline tests multilingual limitations of standard multimodal RAG pipelines trained in high-resource languages. It handles multiple modalities but is monolingual. Queries are machine-translated to SAE, and answers translated back to the target language. Comparing against this baseline highlights that existing multimodal systems cannot transfer knowledge across languages effectively and that machine translation is a poor substitute. This

| Model | Text | | | | Image | | | | Image+Text | | | | All | |
|---|---|---|---|---|---|---|---|---|---|---|---|---|---|---|
| | QA-Acc | P@10 | R@10 | F1@10 | QA-Acc | P@10 | R@10 | F1@10 | QA-Acc | P@10 | R@10 | F1@10 | QA-Acc | F1 |
| Text-Only RAG | 20.6 | 17.3 | 16.9 | 17.1 | 0.5 | 0.5 | 0.4 | 0.4 | 0.2 | 0.8 | 0.6 | 0.7 | 4.6 | 3.8 |
| Monolingual RAG | 22.3 | 18.4 | 17.7 | 18.0 | 12.1 | 10.9 | 11.3 | 11.1 | 12.5 | 10.1 | 9.8 | 9.9 | 18.2 | 12.4 |
| GPT-4o (Translation HRL) | 18.0 | N/A | N/A | N/A | 16.0 | N/A | N/A | N/A | 7.8 | N/A | N/A | N/A | 15.7 | N/A |
| GPT-5 Mini (Translation HRL) | 17.8 | N/A | N/A | N/A | 16.5 | N/A | N/A | N/A | 7.7 | N/A | N/A | N/A | 15.4 | N/A |
| Claude 4.5 Sonnet (Translation HRL) | 18.5 | N/A | N/A | N/A | 17.5 | N/A | N/A | N/A | 8.3 | N/A | N/A | N/A | 17.9 | N/A |
| DeepSeek V3.1 (Translation HRL) | 11.0 | N/A | N/A | N/A | 14.4 | N/A | N/A | N/A | 5.6 | N/A | N/A | N/A | 13.5 | N/A |
| RAGVL + MT | 28.6 | 33.3 | 27.0 | 29.2 | 30.7 | 37.9 | 31.1 | 34.3 | 34.8 | 40.3 | 33.4 | 36.8 | 30.3 | 33.5 |
| GPT-5 Mini + RAG | 26.8 | 30.1 | 24.9 | 27.3 | 28.7 | 33.6 | 28.1 | 30.8 | 32.9 | 35.5 | 29.3 | 32.2 | 28.6 | 30.4 |
| Claude 4.5 Sonnet + RAG | 28.2 | 32.3 | 26.7 | 29.2 | 29.6 | 35.9 | 30.4 | 32.9 | 33.7 | 38.7 | 31.4 | 34.7 | 29.4 | 32.2 |
| DeepSeek V3.1 + RAG | 24.6 | 27.1 | 21.5 | 24.5 | 26.9 | 31.4 | 26.0 | 28.5 | 30.9 | 33.1 | 27.0 | 29.8 | 27.0 | 28.4 |
| Search-R1 + MT | 36.2 | 38.6 | 79.8 | **52.0** | 0.5 | 1.1 | 1.3 | 1.2 | 0.3 | 0.9 | 1.0 | 0.9 | 7.8 | 11.5 |
| XM-RAG | **36.7** | **37.1** | **85.6** | 51.8 | **38.3** | **44.5** | **57.1** | **50.0** | **42.9** | **44.2** | **67.9** | **53.6** | **38.1** | **50.4** |

Table 2: Performance of models on LR-MMQA by query modality.

demonstrates that current models are inadequate for a task that requires multimodal and multilingual capabilities, which is essential for success when evidence in the query language is limited.

**Multimodal RAG Baseline RagVL (Chen et al., 2024)** RagVL is the SOTA on WebQA and MultimodalQA (Full Wiki), combining vision-language modeling with retrieval via knowledge-enhanced reranking and noise-injected training. We adapt it with the same MT process as the multimodal monolingual baseline, showing that even the strongest multimodal RAG systems fail to generalize across languages.

**Frontier MLLMs** We include strong, non-retrieval-augmented baselines representing peak multimodal understanding. Queries are machine translated to english and output is translated back to the target language for evaluation. Despite the capabilities, these models underperform in a real-world, knowledge-intensive setting, illustrating that even advanced commercial models cannot yet handle the challenges of LR-MMQA or difficult knowledge-seeking queries originally posed in low-resource languages.

## 5.3 MAIN RESULTS

**Retrieval Quality Analysis** We evaluate retrieval baselines on our benchmark, which consists of WebQA and MMQA questions where state-of-the-art systems fail, so overall performance is predictably low. For context, full-wiki SOTA F1 on the original WebQA and MMQA is 77.64 and 98.9(Chen et al., 2024), respectively; despite XM-RAGs strong gains below, there remains a large absolute gap. As shown in Table 2, on text questions XM-RAG recall is extremely high (85.6) but paired with the lowest precision (37.1), producing the second lowest F1 across modalities (51.8). This reflects a consistent pattern where XM-RAG recall exceeds precision, stemming from the dense retriever surfacing many cross-lingual candidates that the reranker cannot fully filter. Even so, XM-RAGs and RagVL's higher precision compared with text-only (F1 17.1) and monolingual baselines (18.1) shows the benefit of both baselines reranker and XM-RAG's multilingual retriever, which better prune off-topic candidates and promote more effective search across languages. An example of XM-RAGs successful multi-hop retrieval, compared to baseline failures, appears in Appendix E.

The monolingual baseline outperforms the text-only baseline primarily on image and image+text questions, which make up a large portion of the dataset, while on text-only questions their metrics are close. On both image and image+text queries, XM-RAG substantially outperforms all baselines across metrics (e.g. F1: 50.0 and 53.6), yielding an overall F1 of 50.5, which is 15.3 points higher than the next best baseline (35.2). Although LR-MMQA is built from failure cases, the gap to original WebQA/MMQA highlights persistent limits in multilingual encoding. Addressing the precisionrecall imbalance in retrieval, alongside improving cross-lingual representations, will be essential for future systems to close this disparity.

**Answer Quality** Answer accuracy shows a clear gap between the evaluated systems. GPT-4o outperforms the text-only baseline by 8.5 points, likely because some factual knowledge appears in its training data. The text-only baseline performs near zero on image and multimodal questions (0.4 and 0.2), producing unsupported answers that lower overall accuracy. The monolingual base-

line achieves 9.5 points higher overall accuracy than the text-only baseline by incorporating both text and image evidence. XM-RAG reaches 38.1 accuracy, 6.3 points higher than RagVL and 20.3 points higher than any other baseline. These gains are largely due to its multilingual query encoding and cross-modal fusion, which together retrieve and compress more relevant evidence than the baselines aided by MT.

Although retrieval is often successful, accuracy remains low because even XM-RAG struggles on questions requiring reasoning over multiple highly specific pieces of information (28.7 % of all XM-RAG errors; see Appendix G). Text queries often fail when answers depend on comparing fine-grained details such as dates across documents(45.2%), while image queries often fail when comparing attributes like object colors across multiple images (41.3%). These challenges are amplified in cross-lingual retrieval from low-resource languages, where even small translation errors can prevent precise ground-truth items from being surfaced (31.2% of all failures). This problem is exacerbated by Yoruba questions, where tonal information can be lost in text-only embeddings. For comparison, the full-wiki SOTA accuracy on WebQA and MMQA is 64.40 and 73.48 (Chen et al., 2024), leaving large gaps of 26.3 and 35.4 points. Thus, while XM-RAG sets a new state of the art for LR-MMQA, current systems remain limited in retrieving and reasoning over evidence. Future progress depends on better integration of retrieved context and addressing the additional difficulties posed by low-resource languages and cross-lingual retrieval. Failure examples can be found in Appendix H, I, J

**Qualitative Analysis on Generated Responses** Our qualitative analysis of GPT-4o reveals a consistent geographical bias: queries in Yoruba often yield Nigeria-centered responses, while Tamil queries default to Indian contexts. This behavior highlights the geographical bias inherent in training data, a critical shortcoming in LLMs. Since the majority of training data for LRLs is inherently concentrated in a specific region, the model forms a strong statistical association between the language and its dominant culture.

Such bias is not unique to commercial LLMs, as a monolingual RAG pipeline would face the exact same issue, encountering limited data that only contains information about the country where the language of the query is spoken. On the contrary, The multilingual text-only baseline and XM-RAG did not display the language-culture bias on the same set of questions. Their ability to retrieve multilingually from a wider HRL corpus that transcends single-country limitations allowed them to provide answers that were not confined to a single country or cultural frame, leading to more accurate responses. Reference Appendix D for an example of a question and generated responses falling under this description.

## 5.4 ABLATION STUDY

| Model | QA-Acc | F1 |
|---|---|---|
| XM-RAG (full) | 38.1 | 50.5 |
| w/o Cross-Encoder Reranker | 37.2 | 49.4 |

Table 3: Ablation study comparing XM-RAG with and without the cross-encoder reranker. The best results for each metric are highlighted in bold.

As shown in Table 3, XM-RAG without the cross-encoder reranker achieves an accuracy of 37.2 and a retrieval F1 of only 40.7, a drop of 2.2% compared to the F1 of the full XM-RAG pipeline. As seen in Figure 2, this F1 performance drop occurs consistently in all modalities, highlighting the critical role of the learned reranker in bridging retrieval and reasoning.

The drop in F1 occurs because the reranker contributes to more precise answer grounding: without it, the system tends to select passages or segments that are semantically related but not directly relevant to the query. This results in noisier context, weaker alignment between evidence and the question, and ultimately a degradation in precision, which can be seen in all modalities in Figure 2. Since F1 directly combines both recall and precision, even moderate precision losses cause a decline in overall F1.

Accuracy also declines under this ablation because incorrect or noisy contexts lead the generator to produce answers that are either partially correct or completely off-target. The reranker ensures that

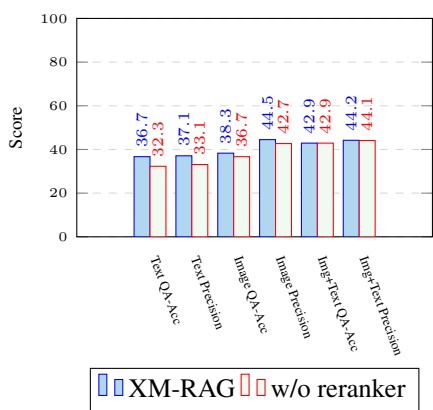

Figure 2: Ablation on reranker: accuracy and precision scores for different modalities.

retrieved evidence is both semantically rich and directly relevant to the query. Without this step, the model's reasoning chain is built on lower-quality foundations, making accurate prediction much less likely.

## 6 CONCLUSION

In this paper, we introduce the first multimodal KBQA dataset for LRLs, LR-MMQA, as well as XM-RAG, a state-of-the-art baseline model for multimodal KBQA in LRLs. LR-MMQA is a benchmark for Tamil and Yoruba that utilizes questions and images from WebQA and MultimodalQA, two open-domain question-answer-answer datasets. We then translated select queries from both datasets into Tamil and Yoruba, containing 718 unique question-answer pairs for each language. We evaluate our baseline model XM-RAG and compare it with existing open-domain benchmark models. Through XM-RAG's unique combination of features, we achieve SOTA metrics across all modalities compared to other baseline models, which we can attribute to XM-RAG's ability to handle both multilingual and multimodal data.

## 7 LIMITATIONS

Due to the absence of a multilingual multimodal RAG model for Tamil and Yoruba for KBQA, there may be limited comparison of XM-RAG to other models. It should be noted that LR-MMQA is a relatively small dataset in comparison to WebQA or MultimodalQA, with a particular emphasis on image questions. In the future, more questions and answers should be created with ground-truth text documents to combat this issue and allow further evaluation. Furthermore, LR-MMQA can also be improved with the inclusion of QA pairs in tonal or highly agglutinative languages to create a more inclusive benchmark and better assess a model's performance across diverse languages. Finally, HRL knowledge bases may not fully reflect low-resource scenarios.

## REPRODUCIBILITY STATEMENT

The code used in this paper can be found here. The steps to reproduce the results are:

1. Clone the repository.
2. Install dependencies using `pip install -r requirements.txt`
3. Download the LR-MMQA benchmark from `https://huggingface.co/datasets/anonymous132145/LR-MMQA`.
4. Download the supporting context images from here.
5. Follow all instructions in `README.md`.

After running the code as outlined in the repository, you should be able to reproduce the evaluation metrics reported in Table 2.

## LLM STATEMENT

LLMs were used in this paper to aid and polish writing and experimental code.

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

## A  TRANSLATION PROMPT

| Yoruba Few-Shot Examples | Tamil Few-Shot Examples |
|---|---|
| **Example 1:**
English: After World War I, a new political landscape emerged in the Middle East.
Yoruba: Lyìn Ogun Àgbáyé Kìíní, ìèlú tuntun kan farahàn ní Àárín Gbùngbùn Il-Ayé.

**Example 2:**
English: The film "Moonlight" won the Academy Award for Best Picture.
Yoruba: Fîìmù "Moonlight" gba Àmì-y Akádmì fún Fîìmù Tó Dára Jù L.

**Example 3:**
English: A well-known saying is "the early bird catches the worm."
Yoruba: Àà àti ìe tí a m jù ni pé "y àár ní mú kòkòrò." | **Example 1:**
English: The police were called to the scene.
Tamil: kaavalthuraiyukku anda idaththirkku vara azhaippu vidukkappattadhu.

**Example 2:**
English: The new species of butterfly was discovered in the rainforest.
Tamil: oru puthiya pattampuuchi inam mazhai kaatil kandupidikkappattadhu.

**Example 3:**
English: The chef prepared a delicious meal using fresh, local ingredients.
Tamil: samayalkaarar puthiya, ulloor porutkalai payanpaduthi oru suvaiyana unavai thayariththaar. |
| **Translation Prompt:**
You are an expert linguist and translator. Your task is to translate a Question-Answer pair from English to the target language. You must maintain the integrity of the question and ensure the translated answer remains a correct, verbatim excerpt from the translated context.

**Instructions:**
1. Translate the question to the target language.
2. Translate the answer to the target language.
3. The translated answer must be a direct, literal substring of the translated text (not paraphrased).
4. Maintain the original format and structure.
5. Ensure all questions and answers are posed as a native speaker would ask and answer.

**Your Task:**
Source Question: {source_question}
Source Answer: {source_answer} | |

Table 4: Few-shot exemplars and translation prompt used for creating LR-MMQA.

# B TRANSLATION EXAMPLES

| English Question | English Answer | Yoruba Question / Answer | Tamil Question / Answer |
|---|---|---|---|
| Who sings the most songs in the world? | Asha Bhosle | Tani o krin plp jùl ní gbogbo àgbáyé? / Asha Bhosle | ulagil adhika paadalgalai paadiyavar yaar? / Aasha Bhosle |
| How many colors are in the Point Skyhawks logo? | 4 | Àwn àw mélòó ló wà nínú àmì Point Skyhawks? / 4 | paayind skaihaaks logo-il etthanai niRangal ullana? / 4 |
| Danish Viking, who ruled over parts of Friesland between 841 and 873, was the uncle of a Viking leader who raided the British Isles, West Francia, Frisia, and Lotharingia in the 860s and 870s? | Roricus, Rorichus | Viking Denmark, tí ó j ba lórí apá kan ti Friesland láàrin dún 841 àti 873 j àbúrò bàbá tàbí ìyá fún olórí Viking kan tí ó klu Erékùù Brítánì, Ìw-òòrùn Francia, Frisia, àti Lotharingia ní grùn-ún dún ksàn-án àti grùn-ún dún kwàá? / Roricus, Rorichus | 841 muthal 873 varai freeslandin pagudigalai aatchi seidha danish viking, 860kal matrum 870kalil british theevugal, merku Francia, frisia matrum lotharingia-kolaiyaditha oru viking thalaivarin maamaa yaar? / Rorikus, Rorichus |

Table 5: Examples of English questions and answers with Yoruba and Tamil translations.

# C SAMPLE PERTURBED DATA POINT

| Field | Content |
|---|---|
| $Q_{EN}$ (**English Question**) | If a partial seizure spreads to the cortex, it can result in what type of tonic-clonic seizure? |
| $Q_{LRL}$ (**Tamil Question**) | paguthi valippu moolaiyin puranukku paravinaal, adhu endha vagaiyana tonic-clonic valippaga maaralaam? |
| $A_{EN}$ (**English Answer**) | Grand mal |
| $A_{LRL}$ (**Tamil Answer**) | grand maal |
| $E_{MM}$ (**Supporting Context**) | N/A |
| $S_{GD}$ (**Ground Truth Documents**) | Seizure types  Wikipedia (title only)
Generalized tonicclonic seizure  Wikipedia (title only)
*Note: Full URLs and snippets omitted for space.* |

Table 6: Example data point from LR-MMQA showing English and Tamil QA pairs, supporting context (if applicable), and gold source titles.

## D  SAMPLE BIASED GPT OUTPUT

| Field | Content |
|---|---|
| $Q_{EN}$ **(English Question)** | Who sings the most songs in the world? |
| $Q_{LRL}$ **(Yoruba Question)** | Ta ni o krin awn orin jù l ni ayé? |
| $A_{GD}$ **(Gold Answer)** | Asha Bhosle |
| $A_{GPT}$ **(GPT Answer)** | Fela Kuti |

Table 7: Example of a biased GPT output where the model incorrectly localized the answer to a Nigerian artist, despite the gold answer being *Asha Bhosle*.

## E  SAMPLE XM-RAG OUTPUT WITH RETRIEVED SOURCES

| Field | Content |
|---|---|
| $Q_{EN}$ **(English Question)** | Which color is found on both the Estonia and Poland Pavilion at Expo 2010? |
| $Q_{LRL}$ **(LRL Question)** | Àw wo ni a rí lórí Ilé Ìgbìm Estonia àti Poland ní Ìfihàn Àgbáyé 2010? |
| $A_{EN}$ **(English Answer)** | Brown is found on both the Estonia and Poland Pavilion at Expo 2010. |
| $A_{LRL}$ **(LRL Answer)** | Àw búráùnì ni a rí lórí Ilé Ìgbìm Estonia àti Poland ní Ìfihàn Àgbáyé 2010. |
| $A_{XM}$ **(XM-RAG Answer)** | Àw búráùnì náá ló wà lórí Ilé Ìgbìm Estonia àti Poland ní Ìfihàn Àgbáyé 2010. |
| $S_{GD}$ **(Gold Sources)** | <ul><li>Estonia Pavilion at Expo 2010 in Shanghai urges action to save the cities.</li><li>Polish Pavilion at Shanghai World Expo 2010.</li></ul> |
| $S_{XM}$ **(Retrieved Sources)** | <ul><li>**Estonia Pavilion at Expo 2010 in Shanghai urges action to save the cities.**</li><li>**Polish Pavilion at Shanghai World Expo 2010.**</li><li>Polish Pavilion / WWA Architects.</li><li>Estonian pavilion for Shanghai EXPO 2010 - Identity.</li><li>Expo 2010 pavilions.</li></ul> |

Table 8: Example XM-RAG successful multi-hop reasoning through answering and retrieval.

## F  TRANSLATION QUALITY EVALUATION

| Language | Sample Size | Adequacy (1-10) | Fluency (1-10) | Inter-Annotator Agreement |
|----------|-------------|-----------------|----------------|---------------------------|
| Tamil | 150 | 8.3 | 8.1 | $\kappa = 0.78$ |
| Yoruba | 150 | 8.1 | 7.9 | $\kappa = 0.74$ |
| Overall | 300 | 8.2 | 8.0 | $\kappa = 0.76$ |

Table 9: Translation quality evaluation results for LR-MMQA dataset. Two native speakers independently assessed translations using 10-point Likert scales for adequacy (semantic correctness) and fluency (naturalness). Inter-annotator agreement measured using Cohen's kappa ($\kappa$).

**Evaluation Protocol:** Two native speakers independently rated each translation on 10-point Likert scales. Disagreements resolved through discussion with a third annotator.

**Rating Scale: Adequacy**: 1 = completely incorrect meaning, 10 = perfect semantic preservation. **Fluency**: 1 = completely unnatural, 10 = native-like naturalness.

## G  FAILURE MODE ANALYSIS

| Failure Type | Text-Only | Image-Only | Image+Text | Overall |
|--------------|-----------|------------|------------|---------|
| Cross-lingual Retrieval | 32.1% | 28.4% | 35.7% | 31.2% |
| Visual Understanding | 0.0% | 41.3% | 29.8% | 25.6% |
| Multi-hop Reasoning | 45.2% | 18.9% | 21.4% | 28.7% |
| Answer Generation | 22.7% | 11.4% | 13.1% | 14.5% |

Table 10: Distribution of failure modes across question types through systematic human categorization of error cases from XM-RAG outputs on LR-MMQA. Authors performed this task.

## H  CROSS-LINGUAL RETRIEVAL FAILURE EXAMPLE

| Field | Content |
|---|---|
| $Q_{EN}$ **(English Question)** | How many people are in the painting of Sappho and Phaon by Jacques-Louis David? |
| $Q_{LRL}$ **(LRL Question)** | Jaak-luuyi Devid varainda Saappo matrum Paayon oviathil eththanai per irukkiraargal? |
| $A_{EN}$ **(English Answer)** | 3 people are in the painting of Sappho and Phaon by Jacques-Louis David. |
| $A_{LRL}$ **(LRL Answer)** | Jaak-luuyi Devid varainda Saappo matrum Paayon oviathil 3 per irukkiraargal. |
| $A_{XM}$ **(XM-RAG Answer)** | Jaak-luuyi Devittin oviangal patri thagaval kidaikkavillai. |
| $S_{GD}$ **(Gold Sources)** | <ul><li>Jacques-Louis David - Sappho and Phaon - WGA6092</li></ul> |
| $S_{XM}$ **(Retrieved Sources)** | <ul><li>Jacques-Louis David paintings overview</li><li>French neoclassical art collection</li><li>David historical paintings</li><li>18th century French artists</li></ul>*Cross-lingual encoding failed to match "Saappo" with "Sappho"* |

Table 11: Example XM-RAG cross-lingual retrieval failure due to semantic drift in proper name encoding.

# I  VISUAL UNDERSTANDING FAILURE EXAMPLE

| Field | Content |
|---|---|
| $Q_{EN}$ **(English Question)** | Looking at Zocalo from Torre Latino Americana how many yellow buildings can be seen? |
| $Q_{LRL}$ **(LRL Question)** | Ni wíwo Zocalo lati Torre Latino Americana, ile pupa melo ni a le ri? |
| $A_{EN}$ **(English Answer)** | Looking at Zocalo from Torre Latino Americana, one yellow building is visible. |
| $A_{LRL}$ **(LRL Answer)** | Ni wíwo Zocalo lati Torre Latino Americana, ile pupa kan ni a le ri. |
| $A_{XM}$ **(XM-RAG Answer)** | Ni wíwo Zocalo lati Torre Latino Americana, awon ile pupo ni a le ri. |
| $S_{GD}$ **(Gold Sources)** | • Zocalo and surroundings as seen from Torre Latinoamericana, Mexico City |
| $S_{XM}$ **(Retrieved Sources)** | • **Zocalo and surroundings as seen from Torre Latinoamericana, Mexico City**
• Mexico City aerial views
• Torre Latinoamericana observation deck
• Historic center Mexico City
• Zocalo plaza architecture |

Table 12: Example XM-RAG visual understanding failure in fine-grained object counting and color identification.

## J   MULTI-HOP REASONING FAILURE EXAMPLE

| Field | Content |
|---|---|
| $Q_{EN}$ **(English Question)** | Which candidate announced their run for president later; Gary Johnson or Michelle Bachmann? |
| $Q_{LRL}$ **(LRL Question)** | Athipar pathavikkaana thangal pottiyai yaar pinthi arivit-thaar; Kaeri Jaansanaa allathu Mishel Baakmanaa? |
| $A_{EN}$ **(English Answer)** | Michelle Baakmanaa |
| $A_{LRL}$ **(LRL Answer)** | Mishel Baakman |
| $A_{XM}$ **(XM-RAG Answer)** | Kaeri Jaansan |
| $S_{GD}$ **(Gold Sources)** | <ul><li>Gary Johnson 2012 presidential campaign - Wikipedia</li><li>Michele Bachmann 2012 presidential campaign - Wikipedia</li></ul> |
| $S_{XM}$ **(Retrieved Sources)** | <ul><li>**Gary Johnson 2012 presidential campaign - Wikipedia**</li><li>**Michele Bachmann 2012 presidential campaign - Wikipedia**</li><li>2012 Republican primary candidates</li><li>Presidential campaign announcements 2011</li><li>Gary Johnson political career</li></ul> |

Table 13: Example XM-RAG multi-hop reasoning failure in temporal comparison synthesis across retrieved documents.

## K   EXPANDED RETRIEVAL BASELINE METRICS AND DEFINITIONS

$$\text{Prec@}k = \frac{|\text{Retrieved}_k \cap \text{Ground Truth Documents}|}{k}$$

$$\text{Rec@}k = \frac{|\text{Retrieved}_k \cap \text{Ground Truth Documents}|}{|\text{Relevant}|}$$

$$\text{F1@}k = 2 \cdot \frac{\text{Prec@}k \cdot \text{Rec@}k}{\text{Prec@}k + \text{Rec@}k}$$

| Model | Text | | | | Image | | | | Image+Text | | | |
|---|---|---|---|---|---|---|---|---|---|---|---|---|
| | **P@2** | **R@2** | **P@5** | **R@5** | **P@2** | **R@2** | **P@5** | **R@5** | **P@2** | **R@2** | **P@5** | **R@5** |
| Text-Only RAG | 17.2 | 16.9 | 17.1 | 16.8 | 0.6 | 0.5 | 0.5 | 0.5 | 0.9 | 0.7 | 0.8 | 0.7 |
| Monolingual RAG | 18.3 | 17.6 | 18.0 | 17.7 | 10.9 | 11.3 | 10.9 | 11.2 | 9.9 | 9.8 | 9.9 | 9.8 |
| RAGVL + MT | 33.4 | 27.3 | 30.6 | 27.9 | 37.8 | 31.6 | 36.4 | 33.2 | 39.6 | 34.2 | 38.1 | 33.7 |
| GPT-5 Mini + RAG | 30.2 | 44.7 | 31.9 | 45.8 | 32.4 | 27.8 | 33.1 | 29.6 | 35.7 | 30.2 | 36.2 | 31.4 |
| Claude 4.5 Sonnet + RAG | 32.1 | 47.6 | 33.3 | 49.8 | 33.8 | 30.2 | 35.1 | 32.4 | 36.2 | 33.4 | 37.0 | 34.2 |
| DeepSeek V3.1 + RAG | 27.3 | 37.9 | 28.6 | 38.7 | 28.2 | 24.3 | 29.7 | 26.1 | 31.7 | 28.2 | 32.9 | 29.6 |
| Search-R1 + MT | 40.1 | 69.7 | 41.6 | 71.8 | 0.6 | 0.8 | 0.9 | 1.1 | 0.5 | 0.9 | 1.0 | 1.2 |
| XM-RAG | **43.4** | **73.8** | **44.5** | **75.6** | **47.6** | **62.1** | **50.8** | **59.2** | **44.0** | **68.8** | **49.2** | **67.3** |

Table 14: Retrieval-only metrics (P@2, R@2, P@5, R@5) for all retrieval baselines on LR-MMQA, split by modality. Translation-to-HRL models excluded.

## L   DE-ANGLICIZED QA PAIR EXAMPLES

| Original English Question | Original Translated Question | English Rewritten Question | Rewritten Question (LRL) | Answer (LRL) |
|---|---|---|---|---|
| How many colors are in the Point Skyhawks logo? | Awọn awò mélòo ló wá nínu ạmí Point Skyhawks? | How many colors are there in the Shooting Stars football team logo? | Awọ mélòo ni o wá nínu ạmi ẹgbé bōọlu Shooting Stars yìi? | 3 (Yoruba) |
| How many colors are in the Point Skyhawks logo? | Paayint Skyhawks logovil ethānai nirangal ullana? | How many colors are there in the Chennai Super Kings team logo? | Chennai Super Kings team logovil ethāna vannam irukku? | 3 (Tamil) |
| What is at the top of the logo for Esporte Clube Santo Andre? | Kí ni ó wá ní òkè ạmí Esporte Clube Santo Andre? | What is at the top of the logo for Crown FC? | Kí ló wá lóri ạmí ıdíje àgbá Crown FC náà? | adé (Yoruba) |
| What is at the top of the logo for Esporte Clube Santo Andre? | Esporte Club Santo Andre-in logo-vin mel dhugalil enna ullathu? | Look at the Chennai Super Kings team logo - what's on it? | Chennai Super Kings ani logo paaru - athoda mela gathula enna irukku? | krīḍam (Tamil) |
| What position are James Brown's hands in? | Ipo wo ni ọwó James Brown wá? | How are Fela Kuti's hands? | Kí ni orúkọ òkè tí orí Fela Kuti wá sí? | ìkúụkú (Yoruba) |
| What position are James Brown's hands in? | James Brown-in kaigal entha nilaiyil ullana? | What position are Rajinikanth's hands? | Rajinikanth kaigal eppadi vachirukkaar? | mushtigal (Tamil) |

Table 15: Examples of de-anglicized QA pairs, showing original English questions, original translated questions, English rewritten questions, rewritten LRL questions, and LRL answers for Yoruba and Tamil. Yoruba uses LaTeX accents for tonal letters. Tamil rewritten questions are fully transliterated.

# M  SINGLE-COMPONENT ABLATIONS BY MODE

| Mode | Component Changed | Swap Variant | P@10 | R@10 |
|---|---|---|---|---|
| 2Text-only | Encoder | Baseline (M-CLIP) | 37.1 | 85.6 |
| | Encoder | BGE | 33.2 | 74.2 |
| 2Text-only | Generator | Baseline (mT5) | 37.1 | 85.6 |
| | Generator | flan-T5-large | 39.2 | 83.1 |
| Image-only | Encoder | Baseline (M-CLIP) | 44.5 | 57.1 |
| | Encoder | BGE | 35.6 | 46.1 |
| Image-only | Generator | Baseline (mT5) | 44.5 | 57.1 |
| | Generator | flan-T5-large | 41.2 | 55.5 |
| Text+Image | Encoder | Baseline (M-CLIP) | 44.2 | 67.9 |
| | Encoder | BGE | 33.7 | 54.3 |
| Text+Image | Generator | Baseline (mT5) | 44.2 | 67.9 |
| | Generator | flan-T5-large | 40.4 | 66.3 |

Table 16: Single-component ablations of XM-RAG, reported separately for each modality mode (text-only, image-only, text+image). Metrics: Precision@10, Recall@10 (retrieval) and Exact Match (EM), token-level F1 (generation).

# N  SAMPLE DATA POINT REQUIRING IMAGE REASONING

| Field | Content |
|---|---|
| $Q_{EN}$ (English Question) | Which religion raises an upraised palm like the Jains do in India and in our cities? |
| $Q_{LRL}$ (Yoruba Question) | sìn wo ló máa  gbé w sókè tí àwn Jains  e ní India àti ní àwn ìlú wa yìí? |
| $A_{EN}$ (English Answer) | Islam |
| $A_{LRL}$ (Yoruba Answer) | Ìsìlámù |
| $E_{MM}$ (Supporting Context) |  |
| **Metadata** | **Question ID:** c81c4000f886ed9ebe29f8989088575e
**Dataset Source:** MMQA
**Modalities:** Text + Image
**Unique LRL Item ID:** c81c4000f886ed9ebe29f8989088575e_yoruba |

Table 17: Example LR-MMQA data point for Yoruba, including rewritten culturally aligned question, translated answer, and multimodal supportive image.

## O EVALUATION OF RE-RANKER (NDCG@10)

| Retrieval Baseline | Text | Image | Image+Text | All |
|---|---|---|---|---|
| XM-RAG | 79.6 | 50.0 | 64.3 | 64.6 |

Table 18: NDCG@10 for retrieval baselines on LR-MMQA across query modalities. Translation-to-HRL models are excluded.

## P XM-RAG PERFORMANCE ON ENGLISH DATASET (MMQA)

| K | Hit@K | P@K | Rec@K | MRR@K | NDCG@K |
|---|---|---|---|---|---|
| **Text Retrieval** | | | | | |
| 5 | 62.8 | 30.3 | 57.5 | 50.0 | 59.2 |
| 10 | 68.7 | 34.0 | 64.4 | 50.8 | 51.6 |
| 20 | 73.3 | 36.8 | 69.9 | 51.1 | 63.0 |
| 50 | 78.7 | 37.1 | 76.1 | 51.3 | 64.3 |
| **Table Retrieval** | | | | | |
| 5 | 94.9 | 39.0 | 94.9 | 87.5 | 89.4 |
| 10 | 96.8 | 39.7 | 96.8 | 87.8 | 90.0 |
| 20 | 98.1 | 41.9 | 98.1 | 87.9 | 90.3 |
| 50 | 99.1 | 42.0 | 99.1 | 87.9 | 90.5 |
| **All Modalities (Text + Table + Image)** | | | | | |
| 5 | 32.3 | 28.4 | 27.2 | 45.7 | 53.9 |
| 10 | 35.3 | 24.6 | 30.0 | 46.1 | 55.0 |
| 20 | 37.6 | 22.5 | 32.1 | 46.2 | 55.6 |
| 50 | 40.4 | 21.1 | 34.2 | 46.3 | 56.1 |

Table 19: Retrieval performance of XM-RAG on the MMQA dataset. Metrics reported: Hit@K, Precision@K, Recall@K, MRR@K, NDCG@K.

| K | EM | F1 |
|---|---|---|
| **Text-only** | | |
| 5 | 29.9 | 35.3 |
| 10 | 27.5 | 31.8 |
| 20 | 27.5 | 31.8 |
| 50 | 27.5 | 31.8 |
| **Table-only** | | |
| 5 | 32.6 | 41.8 |
| 10 | 30.2 | 38.2 |
| 20 | 30.2 | 38.2 |
| 50 | 30.2 | 38.2 |
| **Text + Table + Image Fusion** | | |
| 5 | 30.8 | 35.2 |
| 10 | 26.4 | 32.1 |
| 20 | 26.0 | 30.7 |
| 50 | 26.0 | 30.7 |

Table 20: Generation performance of XM-RAG on the MMQA dataset. Metrics reported: Exact Match (EM) and token-level F1 at different top-K retrieval sizes.

# Q NON-RETRIEVAL BASELINES INCLUDING PROMPT TUNING AND DIRECT INFERENCE

| Model | Text | Image | Image+Text | Overall |
|---|---|---|---|---|
| GPT-4o (Direct LRL Inference) | 5.9 | 7.4 | 3.0 | 6.8 |
| GPT-4o (CoT + Translation  HRL) | 18.5 | 16.2 | 8.0 | 16.0 |
| GPT-4o (Translation  HRL) | 18.0 | 16.0 | 7.8 | 15.7 |
| GPT-5 Mini (Direct LRL Inference) | 6.1 | 7.7 | 3.2 | 7.2 |
| GPT-5 Mini (CoT + Translation  HRL) | 18.2 | 16.8 | 7.9 | 15.7 |
| GPT-5 Mini (Translation  HRL) | 17.8 | 16.5 | 7.7 | 15.4 |
| Claude 4.5 Sonnet (Direct LRL Inference) | 8.2 | 10.4 | 4.1 | 9.9 |
| Claude 4.5 Sonnet (CoT + Translation  HRL) | 19.0 | 17.8 | 8.5 | 18.2 |
| Claude 4.5 Sonnet (Translation  HRL) | 18.5 | 17.5 | 8.3 | 17.9 |
| DeepSeek V3.1 (Direct LRL Inference) | 3.5 | 4.9 | 1.4 | 4.2 |
| DeepSeek V3.1 (CoT + Translation  HRL) | 11.5 | 14.8 | 5.8 | 13.9 |
| DeepSeek V3.1 (Translation  HRL) | 11.0 | 14.4 | 5.6 | 13.5 |

Table 21: Non-retrieval baselines on LR-MMQA. Only QA accuracy is reported per modality and overall. Retrieval metrics are not applicable.

## R  EXPERIMENTAL SETUP AND COMPUTING RESOURCES

The experiments were conducted using a dedicated GPU cluster for training and inference on large models. Below are the specifications and details:

**GPU Resources:** The main experiments were performed on a GPU cluster equipped with 2x NVIDIA A100 SXM GPUs with 251 GB memory each. These GPUs provided high-throughput tensor core acceleration suitable for challenging multimodal and multilingual KBQA tasks.

**CPU Resources:** The cluster included 16 vCPUs, which were used for data preprocessing, baseline evaluations, and lightweight model inference tasks alongside GPU computations.

**Memory:** The GPU cluster had sufficient system RAM to manage large datasets and multimodal inputs efficiently. The 251 GB GPU memory per card allowed for batch processing and minimized data offloading during model execution.

**Storage:** Experiments utilized high-speed SSD storage on the cluster to handle the 1,436 KBQA examples from the benchmark, including multimodal inputs such as images and structured knowledge representations.

**Experiment Details:**

- **RAMQA and SKURG:** Running RAMQA and SKURG on the benchmark took approximately 5 hours for WebQA and 3 hours for MultimodalQA. These times reflect the complexity of reasoning across multiple hops and modalities.
- **Baseline Models:** Running all other baseline models on the same benchmark is estimated to take an additional 2-4 hours each, considering the relatively small dataset size (1,436 examples) but challenging multimodal multi-hop questions. This estimate accounts for per-sample inference times, preprocessing overhead, and model loading times on the cluster.

**Total Computing Time:** In total, including running RAMQA, SKURG, and all baseline models, the experiments required roughly 10-15 GPU hours and approximately 20-25 CPU hours on the cluster for preprocessing and supporting tasks. This configuration ensured that all models could be executed efficiently while handling the high memory and computational demands of multimodal KBQA reasoning tasks.

