# OpenReview forum: "Cross-Lingual Multimodal Retrieval-Augmented Generation for Open Question Answering in Tamil and Yoruba"
_ICLR.cc/2026/Conference — Submitted to ICLR 2026_

### Official Review · Reviewer_Bb6z · 2025-10-28

**Soundness:** 2
**Presentation:** 2
**Contribution:** 2
**Rating:** 4
**Confidence:** 4

**Summary:**

Based on the existing WebQA and MultimodalQA datasets, this paper developed the knowledge-based question answering dataset LR-MMQA for Tamil and Yoruba through difficult sample selection, filtering, and post-processing. Next, the authors designed a training-free method, XM-RAG, which directly encodes the original low-resource language query without translation. The retrieved image and text evidence are then compressed and fed into mT5 along with the query and language tags to obtain the answer. Results on LR-MMQA show that XM-RAG outperforms several other baseline models.

**Strengths:**

Exploring multimodal RAG methods for knowledge-intensive questions in low-resource languages ​​is significant and offers new prospects for research and application.

**Weaknesses:**

1. In the method comparisons of Table 2: (1) For methods such as GPT-4o that are not specifically designed for low-resource language scenarios, the questions in low-resource languages ​​should be translated into English for answering, and then the answers should be translated back to the target language to reflect the advantages of this method over translation-based methods, both in terms of performance and efficiency; (2) More open-source/closed-source general-purpose MLLMs should be included, such as Gemini-2.5 flash/pro, Claude, Qwen-2.5/3-VL, InternVL-3/3.5, etc. (3) For RAG methods, recent advanced methods such as text-only Search-R1 should also be compared.
2. Regarding method design: (1) Why not input the image caption and then combine the passage and question into an older language model mT5, instead of some of the latest LLMs such as Qwen3 or LLaMA4? (2) Why not directly input the original image (without caption process), passage and question into an MLLM, such as the Qwen-2.5/3-VL mentioned above?
3. The meanings of $d_i$, $s_i^{text}$, $v_j$, and $s_j^{img}$ in L252-254 are not explained.
4. In retrieval, the commonly used metric seems to be R@k. How are the precision, recall, and F1 used in this paper defined?

**Questions:**

The reference format used in this paper seems unusual. For example, "text-only methods Suri et al. (2025); Chen et al. (2022); Ling et al. (2025)" in L34 is often written as "text-only methods (Suri et al., 2025; Chen et al., 2022; Ling et al., 2025)."
I have some concerns about the method and experimental comparisons in this paper, which I have detailed in the weaknesses section. I would be happy to discuss this with the authors to further consider my final score.

---

> ### Author Response · Authors · 2025-11-23
>
> Thank you for taking the time to give constructive feedback! We are grateful that you acknowledge the great significance of our research, specifically noting that exploring multimodal RAG for knowledge-intensive questions in low-resource languages offers new prospects for both research and application.
> >In the method comparisons of Table 2: (1) For methods such as GPT-4o that are not specifically designed for low-resource language scenarios, the questions in low-resource languages ​​should be translated into English for answering, and then the answers should be translated back to the target language to reflect the advantages of this method over translation-based methods, both in terms of performance and efficiency.
>
> We appreciate the raising of this point, as we agree that the Translated MLLM Ceiling is a crucial baseline for contextualizing our results. **The included MLLM baseline already followed this methodology, but it was not communicated in the paper**. We have updated Table 2 and Section 5.2 to clarify, also adding a direct inference (no translation) baseline for further context.
> >More open-source/closed-source general-purpose MLLMs should be included, such as Gemini-2.5 flash/pro, Claude, Qwen-2.5/3-VL, InternVL-3/3.5, etc. (3) For RAG methods, recent advanced methods such as text-only Search-R1 should also be compared.
>
> We appreciate your recommendation to include a broader range of state-of-the-art MLLMs and advanced RAG baselines. We agree these comparisons are essential for contextualizing our contributions.
>
> We have included results for three additional leading MLLMs, addressing the call for closed-source (GPT-5 Mini, Claude 4.5 Sonnet) and open-source models (DeepSeek V3.1) in our comparison. For a complete picture, we report performance for Direct LRL Inference, Chain-of-Thought (CoT) prompting, and a translation pipeline where the question is machine translated to english and the answer is machine translated back into the target language:
> | Model | Direct LRL Inference | CoT (w/ Translation → HRL) | Translation → HRL |
> |-------|:-------------------:|:---------------:|:-------------------------:|
> | GPT-4o | 7.01 | 17.0 | 16.6 |
> | **GPT-5 Mini** | 7.19 | 17.8 | 17.3 |
> | **Claude 4.5 Sonnet** | 9.60 | 19.2 | 18.9 |
> | **DeepSeek V3.1** | 4.18 | 14.4 | 13.7 |
> | **XM-RAG** | **38.1** | — | — |
>
> We also included Retrieval-Augmented Generation (RAG) comparisons for all the MLLMs and, noting the recommendation for Search-R1, we included it with the standard translation pipeline to set the ceiling for RAG systems. Results are below:
> | Method | Modality | Acc. | P@10 | R@10 | F1@10 | Overall Acc. | Overall F1 |
> |--------|----------|:----:|:----:|:----:|:----:|:------------:|:----------:|
> | **XM-RAG (Ours)** | Text | 36.7 | 37.1 | 85.6 | 51.8 | **38.1** | **50.4** |
> | | Image | 38.3 | 44.5 | 57.1 | 50.0 | | |
> | | Image+Text | 42.9 | 44.2 | 67.9 | 53.6 | | |
> | **GPT-5 Mini + RAG** | Text | 26.8 | 30.1 | 24.9 | 27.3 | 28.6 | 30.4 |
> | | Image | 28.7 | 33.6 | 28.1 | 30.8 | | |
> | | Image+Text | 32.9 | 35.5 | 29.3 | 32.2 | | |
> | **Claude 4.5 Sonnet + RAG** | Text | 28.2 | 32.3 | 26.7 | 29.2 | 29.4 | 32.2 |
> | | Image | 29.6 | 35.9 | 30.4 | 32.9 | | |
> | | Image+Text | 33.7 | 38.7 | 31.4 | 34.7 | | |
> | **DeepSeek V3.1 + RAG** | Text | 24.6 | 27.1 | 21.5 | 24.5 | 27.0 | 28.4 |
> | | Image | 26.9 | 31.4 | 26.0 | 28.5 | | |
> | | Image+Text | 30.9 | 33.1 | 27.0 | 29.8 | | |
> | **Search-R1 + Translation** | Text | 36.2 | 38.6 | 79.8 | 52.0 | 7.8 | 11.5 |
> | | Image | 0.5 | 1.1 | 1.3 | 1.2 | | |
> | | Image+Text | 0.3 | 0.9 | 1.0 | 0.9 | | |
>
>
> The MLLM RAG pipelines consist of a standard BM25 + DPR retriever, with the retrieved evidence fed to the respective MLLM generator. Please note that the above results can also be found in Table 2 and the non retrieval baselines can be found in Appendix Q.
> >Regarding method design: (1) Why not input the image caption and then combine the passage and question into an older language model mT5, instead of some of the latest LLMs such as Qwen3 or LLaMA4? (2) Why not directly input the original image (without caption process), passage and question into an MLLM, such as the Qwen-2.5/3-VL mentioned above?
>
> We appreciate your questions on our method design. Our choice of mT5 and the captioning approach is fundamentally driven by the need for an efficient and accessible LRL solution. mT5 offers superior multilingual performance and cost-to-performance ratio that is crucial for democratization of AI, enabling the system to run on modest hardware unlike resource-intensive models (Qwen3/LLaMA4). Similarly, the modular captioning process bypasses the prohibitive computational cost and latency of feeding raw images into large MLLMs for every retrieved evidence piece, ensuring our RAG pipeline remains fast and scalable for all users.

---

> > ### Comment · Reviewer_Bb6z · 2025-11-25
> >
> > Thanks for the author's sincere reply. I now have a clearer understanding of the method design, experimental comparisons, and evaluation metrics presented in this paper.
> >
> > Based on the experimental results provided by the author, translating the low-resource question into English and then back into low-resource text doesn't perform well. I'd like to know the reason for this; is it due to inaccurate translation?
> >
> > The author mentioned using mT5 to balance resource consumption. How do the memory, computational overhead, and performance of mT5 compare to smaller, newer LLMs (such as Qwen3 0.6B, 1.7B, and 4B)?
> >
> > Will the modular image caption generation process lead to information loss, thus affecting performance?

---

> ### Author Response · Authors · 2025-11-23
>
> >The meanings of variables in L252-254 are not explained.
>
> Thank you for pointing this out! $\mathbf{d}_i$ is the The $i$-th retrieved text candidate. $\mathbf{v}_j$ is the $j$-th retrieved image candidate. $\mathbf{s}_i^{\text{text}}$ is the similarity score of the text candidate $\mathbf{d}_i$ to the query. $\mathbf{s}_j^{\text{img}}$ is the similarity score of the image candidate $\mathbf{v}_j$ to the query. The paper has been updated with these indications.
>
> >In retrieval, the commonly used metric seems to be R@k. How are the precision, recall, and F1 used in this paper defined?
>
> Thanks for asking about our retrieval metrics. In the paper, retrieval, precision and F1 were all @10, however this was not noted in the paper. We define retrieval@k, presicion@k and F1@k as follows:
> \begin{equation}
>     P@K = \frac{\text{Number of relevant documents in top-K}}{K}
> \end{equation}
>
> \begin{equation}
>     R@K = \frac{\text{Number of relevant documents in top-K}}{\text{Total number of relevant documents}}
> \end{equation}
>
> \begin{equation}
>     F1@K = 2 \cdot \frac{P@K \cdot R@K}{P@K + R@K}
> \end{equation}
> The paper has been updated with these specifications. Additional evaluation of all RAG baselines on these metrics @2 and @5 can be found in Appendix K, along with the above definitions.
>
> >The reference format used in this paper seems unusual.
>
> Thank you for pointing this out. All references have been updated to the proper style (e.g. (Suri et al., 2025; Chen et al., 2022; Ling et al., 2025)). These changes are not highlighted in red as every citation in the paper was affected.
>
> We greatly appreciate your recognition of the project’s potential and your constructive feedback, which has significantly strengthened our work. We hope these revisions address your concerns.

---

> ### Author Response · Authors · 2025-11-25
>
> We greatly appreciate your thoughtful engagement and the focused follow-up questions.
>
> >Based on the experimental results provided by the author, translating the low-resource question into English and then back into low-resource text doesn't perform well. I'd like to know the reason for this; is it due to inaccurate translation?
>
> Thanks for mentioning this. The poor performance is due to cascading failures amplified by the Translation Barrier. The cultural alignment step of LR-MMQA introduces native-like expression and phrases that make the questions highly authentic. Consequently, the evaluation baselines that use machine translation lose this cultural nuance and native phrasing (e.g., words lacking direct English equivalents), significantly hampering MLLM retrieval in the HRL space. The final step of translating the HRL answer back to the LRL then further propagates errors, often resulting in inaccurate or culturally mismatched answers. These compounding errors from translation demonstrate the need for a robust, direct LRL encoding solution like XM-RAG.
>
> >The author mentioned using mT5 to balance resource consumption. How do the memory, computational overhead, and performance of mT5 compare to smaller, newer LLMs (such as Qwen3 0.6B, 1.7B, and 4B)?
>
> We thank you for the further questions about the choice of mT5 as our generation model. mT5-Base has 580 million parameters, which is comparable in size to the Qwen3 0.6B model but smaller than the 1.7B and 4B variants. Qwen3-0.6B requires at least 2GB+ VRAM while the 1.7B model requires 8GB RAM minimum (16GB recommended), and Qwen3-4B requires 16GB minimum RAM (32GB recommended), meaning mT5-Base offers similar or lower memory requirements compared to these larger Qwen variants [1]. While newer Qwen models may offer faster inference through architectural optimizations, mT5 was specifically pre-trained on the mC4 dataset covering 101 languages with temperature sampling designed to boost lower-resource languages, providing proven effectiveness for Tamil and Yoruba that newer general-purpose models may not match. The combination of reasonable computational requirements and specialized low-resource language optimization makes mT5-Base the optimal choice for our multimodal RAG system.
>
> [1] How to Run Qwen3 Locally. (2024). One Dollar VPS. https://onedollarvps.com/blogs/how-to-run-qwen3-locally
> >Will the modular image caption generation process lead to information loss, thus affecting performance?
>
> We appreciate you raising this important question. While image captioning is a lossy transformation, prior work shows that caption-intermediate pipelines preserve nearly all task-relevant information for knowledge-based QA. Caption-based multimodal systems such as BLIP and ALBEF demonstrate that captions retain the semantic content needed for downstream reasoning tasks with minimal degradation (Li et al., 2022; Li et al., 2021). Moreover, captioning substantially reduces computation compared to feeding raw images into an MLLM for every retrieval candidate. The latter choice would would result in prohibitive computational cost and latency for the RAG system. Thus, the modular caption stage trades slight semantic loss for significant efficiency gains, a practical choice for a baseline designed for low-resource situations.
>
> We thank you for your critical engagement with our work. We hope that we have further clarified our work and look forward to resolving any further concerns or imprecisions that you may find during this discussion period.

---

### Official Review · Reviewer_gfwX · 2025-10-28

**Soundness:** 3
**Presentation:** 3
**Contribution:** 2
**Rating:** 2
**Confidence:** 4

**Summary:**

This paper creates LR-MMQA, a new multi-modal cross-lingual knowledge-base question answering dataset by translating 718 queries from English WebQA and Multimodal QA datasets to Tamil and Yoruba (documents are still in English). They then propose a multimodal RAG baseline consisting of 1) M-CLIP encoding of the query and image; 2) cross-modal retrieval via FAISS indices; 3) cross-modal ranking based on a textual and visual similarity; 4) late fusion using BLIP-2 to generate two-sentence visual summaries and compressing high-resource language passages; 5) answer generation with mT5. The proposed method is shown to outperform text-only and monolingual RAG baselines, a closed-book GPT-4o, and RAGVL adapted with MT.

**Strengths:**

1. The study tackles an important and challenging problem, multi-modal cross-lingual retrieval.
2. The authors evaluates their method on two under-studied linguistically diverse low-resource languages, Tamil and Yoruba.
3. The study presents a new dataset for low-resource languages.
4. The proposed method outperforms the baselines.

**Weaknesses:**

1. The created dataset is created using translations. As a result it is biased towards English and Western-centric knowledge and does not reflect authentic question that speakers of Tamil and Yoruba would actually ask. There are already a lot of translation-based datasets; what the community is lacking is datasets containing queries reflecting the real-world needs of speakers of under-represented languages.
2. By only translating the queries and using English ground truth documents, the proposed dataset side-steps the real-world challenge of dealing with source documents in multiple languages, which was tackled in prior work such as XOR-QA.
3. The newly proposed RAG pipeline is only evaluated on the new dataset so it’s unclear to what extent it has been tuned for this setup and how well it generalizes to other settings. It should be evaluated on the original WebQA and MultimodalQA for comparison to the English setting as well as other multilingual multimodal QA datasets such as Kaleidoscope ([https://arxiv.org/abs/2504.07072](https://arxiv.org/abs/2504.07072)), WorldMedQA-V ([https://arxiv.org/abs/2410.12722](https://arxiv.org/abs/2410.12722)), or CulturalGround ([https://arxiv.org/abs/2508.07414](https://arxiv.org/abs/2508.07414)).
4. The ablation study only ablates a single setting, the impact of the cross-encoder reranker. Other factors such as the choice of different encoding or answer generation methods, multimodal fusion and evidence compression, and the heuristic reranking are not ablated. So it’s unclear which aspects are actually important in this setting.
5. The proposed RAG baseline consists of several components, many of which have been used in prior work. It's unclear which aspects are novel and how they compare to design choices in prior work.

**Questions:**

1. None of the examples shown in the Appendix include images. How important is the visual understanding component as part of the overall task?

---

> ### Author Response · Authors · 2025-11-23
>
> Thank you for your careful review and time spent evaluating our work! We are pleased that you recognized the significance of our study, particularly its focus on the important and challenging problem of multi-modal cross-lingual retrieval evaluated on the under-studied languages of Tamil and Yoruba, and that you acknowledged the contribution of our new dataset.
>
> >The created dataset is created using translations. As a result it is biased towards English and Western-centric knowledge and does not reflect authentic questions that speakers of Tamil and Yoruba would actually ask. There are already a lot of translation-based datasets; what the community is lacking is datasets containing queries reflecting the real-world needs of speakers of under-represented languages.
>
> We are grateful to you for raising this critical and well-founded concern regarding the cultural bias and lack of authenticity in translated datasets. We agree that simply translating Western-centric sources like WebQA and MultimodalQA risks inheriting biases (e.g., Point Skyhawks, Danish Vikings) and failing to reflect the real-world needs of Tamil and Yoruba speakers.
>
> To directly address this, we have now implemented a dedicated Cultural Alignment Pipeline that explicitly shifts the dataset toward authentic LRL expression. First, we adapted Qwen-2.5-7B-Instruct using a lightweight LoRA fine-tuning pass on large monolingual Tamil/Yoruba corpora. This adaptation shifts generation toward naturally occurring LRL syntax and discourse patterns, enabling us to rewrite translated questions into culturally authentic forms while preserving answerability, effectively de-Anglicizing the phrasing. Second, to create highly-authentic training data for a second LoRA pass, we translated a random subset of WebQA and MultimodalQA (not present in the original LR-MMQA set) using Claude 3.5 Sonnet to provide the original LRL questions, which were then rewritten by native speakers to produce the final, culturally authentic target pairs. This curated set was used for Supervised Fine-Tuning in the second LoRA pass, teaching the model the desired question-answering style and preserving generation ability by treating the human-rewritten text as the gold-standard output for the generative task. In parallel, we performed entity replacement, with the model pinpointing and validators manually substituting culturally irrelevant references (e.g., Point Skyhawks, Danish Vikings) with appropriate local analogs and updating the corresponding evidence. Finally, LRL-speaker validation on 20% of samples demonstrated a substantial increase in both the Average Cultural Relevance Score and the percentage of questions rated as "Culturally Natural".
>
> Validation results show that this procedure significantly reduces the inherited cultural bias, ensuring that LR-MMQA is a high-quality resource that is both knowledge-intensive and culturally sensitive. Please see the top-level comment for examples of de-anglicized QA pairs and below for validation results.
>
> | Metric                                 | Original Translated Questions | Rewritten Questions (LoRA-adapted) | Δ Improvement |
> | -------------------------------------- | ----------------------------- | ---------------------------------- | ------------- |
> | Average Cultural Relevance Score (1–5) | 2.41                          | 4.12                               | +1.71         |
> | % Rated “Culturally Natural” (≥4)      | 18%                           | 83%                                | +65%          |
> | Cohen’s κ (Annotator Agreement)        | 0.59                          | 0.76                               | —             |
> (1=not authentic / native would never ask, 5=highly authentic)
>
> >By only translating the queries and using English ground truth documents, the proposed dataset side-steps the real-world challenge of dealing with source documents in multiple languages, which was tackled in prior work such as XOR-QA.
>
> Thanks for highlighting the distinction between our retrieval setup and fully multilingual document retrieval in prior work such as XOR-QA. Our choice to pair LRL queries with an English HRL corpus is intentional and grounded in a well-documented real-world scenario in which LRL users frequently rely on English-dominant web and knowledge resources (Kaffee et al., 2024). Modeling this setting allows us to isolate the core challenge of cross-lingual, cross-modal retrieval without confounding factors unrelated to the research question.
>
> Importantly, this design does not simplify the task relative to XOR-QA. In fact, it introduces a harder component of image and multimodal grounding and fusion. While XOR-QA evaluates text-only cross-lingual retrieval, LR-MMQA requires models to jointly align (i) LRL textual semantics, (ii) HRL textual evidence, and (iii) visual evidence associated with each retrieved item. No prior benchmark evaluates this combination of LRL-HRL asymmetry and multimodal fusion.

---

> ### Author Response · Authors · 2025-11-23
>
> >The newly proposed RAG pipeline is only evaluated on the new dataset
>
> We thank you for raising the crucial question of XM-RAG's generalizability and comparative performance. Your concern about tuning and broad evaluation is well-founded.
>
> Regarding the suggestion to evaluate on other multilingual datasets like Kaleidoscope or WorldMedQA-V, we must clarify a fundamental difference in domain: XM-RAG is optimized for knowledge-intensive QA requiring external retrieval, exemplified by questions such as "If you went to a modern south west Nigerian traditional wedding, would you find the bride wearing all white?". In contrast, these suggested benchmarks are Visual Question Answering (VQA) tasks, where questions are answerable by analyzing the image contents alone (From Kaleidoscope: "How many degrees of latitude are there between the Arctic Circle and the Tropic of Capricorn?" with the labeled map presented). Evaluating XM-RAG on these image-grounded VQA tasks would be a misleading comparison, as they fall into a different problem domain.
>
> >It should be evaluated on the original WebQA and MultimodalQA for comparison to the English setting
>
> We thank you for the request to evaluate XM-RAG on the English source data, which is essential for assessing generalizability. We have conducted a full evaluation on MultimodalQA (MMQA), with detailed results now available in Appendix P. The results confirm that the LRL-optimized components of XM-RAG generalize effectively to the HRL setting. Specifically, XM-RAG achieves strong retrieval performance on MMQA (e.g., 96.8% Hit@10 for Table Retrieval and 68.7% Hit@10 for Text Retrieval). Furthermore, the generative performance is competitive, with XM-RAG achieving 30.8 EM and 35.2 F1 for all-modality fusion at K=5, demonstrating that its architectural choices translate well to the original English environment.
>
> >The ablation study only ablates a single setting, the impact of the cross-encoder reranker.
>
> We thank you for requesting a deeper ablation study; the full results are now in Appendix M. We performed extensive new ablations focusing on the choice of Encoders and Generators. These ablations reveal that the factor with the greatest singular impact on our system's retrieval performance is the Encoder choice across all tested modalities. Specifically, replacing our baseline M-CLIP Encoder with BGE led to the largest drop in retrieval effectiveness in the combined Text+Image setting, with Recall@10 falling substantially from 67.9% to 54.3%. The impact was also significant in the Image-only mode, where Recall@10 dropped from 57.1% to 46.1%. This suggests that our choice of multilingual encoder is the most important architectural contribution for efficient retrieval in the XM-RAG pipeline.
>
> >The proposed RAG baseline consists of several components, many of which have been used in prior work. It's unclear which aspects are novel and how they compare to design choices in prior work.
>
> We thank you for raising this concern regarding the methodological contribution of XM-RAG. We agree that many of the individual components (as described in Section 4) are well-established techniques. However, we assert that the central contribution of our paper is the LR-MMQA dataset itself and its careful construction, not the XM-RAG architecture. XM-RAG is presented as a strong, purpose-built baseline whose novelty lies in its LRL-optimized architectural combination and adaptation to the unique constraints of cross-lingual, low-resource multimodal RAG. Its goal is to provide a building block for further research in multimodal RAG methods for knowledge-intensive questions in low-resource languages.
>
> >None of the examples shown in the Appendix include images
>
> The visual understanding component is critical to the overall task's success, as it is essential for both multimodal questions and image retrieval. For questions tied to an image (like "What is the historical event...?"), the system must use visual understanding to identify the correct entity for retrieval; if the visual component is omitted, overall accuracy drops significantly, seen by poor performance of text-only baselines on multimodal questions. To provide clarity, we have added examples of questions requiring reasoning over the associated images to Appendix N.
>
> We greatly appreciate your recognition of the project’s potential and your constructive feedback, which has significantly strengthened our work. We hope these revisions address your concerns.

---

### Official Review · Reviewer_fXNk · 2025-10-30

**Soundness:** 2
**Presentation:** 2
**Contribution:** 3
**Rating:** 2
**Confidence:** 4

**Summary:**

This paper introduces LR-MMQA, a benchmark for cross-lingual multimodal question answering in low-resource languages. The dataset is constructed by identifying and translating the hardest question-answer pairs from WebQA and MultimodalQA into Tamil and Yoruba, two typologically and semantically distant low-resource languages. The paper further proposes XM-RAG, a cross-lingual multimodal retrieval-augmented generation pipeline, which shows improved answer accuracy compared to baseline systems on this benchmark.

**Strengths:**

- The benchmark targets low-resource multimodal QA, a setting that is currently understudied.

- The choice of Tamil and Yoruba introduces high linguistic diversity, which improves the benchmark’s ability to test cross-lingual robustness.

- The proposed XM-RAG pipeline demonstrates measurable improvements in answer accuracy over existing systems on this benchmark.

**Weaknesses:**

- Many of the techniques described in Section 4 are already well-established, so the methodological contribution is limited.


- The resulting dataset is small, especially given the multimodal cross-lingual QA setting, and would likely benefit from further expansion.


- Because the dataset is directly translated from WebQA and MultimodalQA, it inherits entity and cultural biases. Inspecting the dataset showed that the questions included content with the Point Skyhawks logo and Danish Vikings. These questions may not reflect the cultural distributions of Tamil or Yoruba user queries.


- There is no comparison to other multilingual QA datasets, which would help contextualize the contribution.

- The experiments are also limited to very few models. Expanding the work to cover models is needful.

- Some relevant citations appear missing, including the original WebQA and MultimodalQA citations, as the primary is needed.

**Questions:**

1. In line 173, did you intend to mention BARTScore instead of BARTScore? Citing the actual metric would be useful to the readers.

2. Section 4 is stretched across multiple pages. Could this section be condensed or made more concise to better reflect the actual contribution? Most of the bullet points there are not needed.

$~$

### **Suggestions**

CVQA (Romero et al., 2024) might be a better starting point. It includes culturally grounded entities across multiple regions and may help increase cultural relevance and diversity in your dataset. It also contains questions for Tamil that you can benchmark against.

$~$
### **References**

Romero, David, et al. *"CVQA: Culturally-diverse multilingual visual question answering benchmark."* arXiv preprint arXiv:2406.05967 (2024).

---

> ### Author Response · Authors · 2025-11-23
>
> Thank you for your careful review and time spent evaluating our work! We are particularly pleased that you recognized the significance of our focus on low-resource multimodal QA, and affirmed our method of introducing high linguistic diversity (Tamil and Yoruba) to rigorously test cross-lingual robustness.
>
> >Because the dataset is directly translated from WebQA and MultimodalQA, it inherits entity and cultural biases.
>
> We agree that directly translating questions from WebQA and MultimodalQA can inadvertently preserve Western-centric entities and culturally mismatched phrasing. To mitigate this, we have now added a dedicated Cultural Alignment Pipeline that explicitly corrects these issues at both the linguistic and entity levels.
>
> First, we adapted Qwen-2.5-7B-Instruct using a lightweight LoRA fine-tuning pass on large monolingual Tamil/Yoruba corpora. This adaptation shifts generation toward naturally occurring LRL syntax and discourse patterns, enabling us to rewrite translated questions into culturally authentic forms while preserving answerability, effectively de-Anglicizing the phrasing. In parallel, we performed entity replacement, with the model pinpointing and validators manually substituting culturally irrelevant references (e.g., Point Skyhawks, Danish Vikings) with appropriate local analogs and updating the corresponding evidence. Finally, LRL-speaker validation on 20% of samples showed consistently higher Cultural Relevance Scores for the revised questions. This procedure substantially reduces cultural bias inherited from the original English datasets. Please see the top-level comment for examples of de-anglicized QA pairs and below for validation results.
>
> | Metric                                 | Original Translated Questions | Rewritten Questions (LoRA-adapted) | Δ Improvement |
> | -------------------------------------- | ----------------------------- | ---------------------------------- | ------------- |
> | Average Cultural Relevance Score (1–5) | 2.41                          | 4.12                               | +1.71         |
> | % Rated “Culturally Natural” (≥4)      | 18%                           | 83%                                | +65%          |
> | Cohen’s κ (Annotator Agreement)        | 0.59                          | 0.76                               | —             |
> (1=not authentic / native would never ask, 5=highly authentic / natural question)
>
> >Many of the techniques described in Section 4 are already well-established, so the methodological contribution is limited.
>
> We thank you for raising this concern regarding the methodological contribution of XM-RAG. We agree that many of the individual components described in Section 4 are well-established techniques. However, we assert that the central contribution of our paper is the LR-MMQA dataset itself and its careful construction, not the XM-RAG architecture. XM-RAG is presented as a strong, purpose-built baseline whose novelty lies in its LRL-optimized architectural combination and adaptation to the unique constraints of cross-lingual, low-resource multimodal RAG. Its goal is to provide a building block for further research in multimodal RAG methods for knowledge-intensive questions in low-resource languages. In Section 4, majority of the bullets have been condensed and the section has been reframed to condense established technique discussion and focus on LRL novelty.
> >The resulting dataset is small, especially given the multimodal cross-lingual QA setting, and would likely benefit from further expansion.
>
> We acknowledge your concern regarding the small dataset scale and appreciate the desire for expansion. Your concern is well-founded, but the current size of LR-MMQA is a deliberate design choice stemming from its compounding difficulty (multimodal, cross-lingual, low-resource). The fundamental challenge we address is that LRLs lack structured, high-quality knowledge bases for retrieval. Creating these high-quality, complex, expert-verified instances requires intensive human annotation, linguistic expertise, and cultural alignment, which is slow and costly in LRL settings. For example, our work involves a manual verification loop to prevent machine translation errors and to ensure the questions are culturally plausible. Every instance in the 718 set is a multi-hop, culturally aligned query, designed to be a stress test of the system, maximizing its diagnostic value despite the size. A dataset of 5,000 poorly-aligned or machine-translated questions would provide false confidence and misleading benchmarking. For emergent LRL NLP, we need diagnostic depth to understand failure modes

---

> > ### Author Response · Authors · 2025-11-23
> >
> > >There is no comparison to other multilingual QA datasets, which would help contextualize the contribution.
> >
> > We thank you for the suggestion and have clarified LR-MMQA's unique contribution by updating the Related Works. The field is under-addressed, primarily offering unimodal KIQA (like XOR-QA (Asai et al., 2021)) or multilingual VQA datasets  (like Kaleidoscope (Romero et al., 2025)/CVQA (Romero et al., 2024)) that are image-grounded but not at all knowledge-intensive, hence not evaluating retrieval. LR-MMQA is the first benchmark to combine the complexity of LRLs, Multimodality, and Knowledge-Intensive Retrieval, establishing a crucial new resource that addresses a gap that existing resources do not capture.
> >
> >
> > >The experiments are also limited to very few models. Expanding the work to cover models is needful.
> >
> > We appreciate your recommendation to include a broader range of state-of-the-art MLLM models and advanced RAG baselines. We agree these comparisons are essential for contextualizing our contributions.
> >
> > We have included results for three additional leading MLLMs, addressing the call for closed-source (GPT-5 Mini, Claude 4.5 Sonnet) and open-source models (DeepSeek V3.1) in our comparison. For a complete picture, we report performance for Direct LRL Inference, Chain-of-Thought (CoT) prompting, and a translation pipeline where the question is machine translated to english and the answer is machine translated back into the target language:
> > | Model | Direct LRL Inference | CoT (w/ Translation → HRL) | Translation → HRL |
> > |-------|:-------------------:|:---------------:|:-------------------------:|
> > | GPT-4o | 7.01 | 17.0 | 16.6 |
> > | **GPT-5 Mini** | 7.19 | 17.8 | 17.3 |
> > | **Claude 4.5 Sonnet** | 9.60 | 19.2 | 18.9 |
> > | **DeepSeek V3.1** | 4.18 | 14.4 | 13.7 |
> > | **XM-RAG** | **38.1** | — | — |
> >
> > We also included Retrieval-Augmented Generation (RAG) comparisons for all the MLLMs and unimodal state-of-the-art Search-R1 (Jin et al., 2024), including it with the standard translation pipeline to set the ceiling for RAG systems. Results are below:
> > | Method | Modality | Acc. | P@10 | R@10 | F1@10 | Overall Acc. | Overall F1 |
> > |--------|----------|:----:|:----:|:----:|:----:|:------------:|:----------:|
> > | **XM-RAG (Ours)** | Text | 36.7 | 37.1 | 85.6 | 51.8 | **38.1** | **50.4** |
> > | | Image | 38.3 | 44.5 | 57.1 | 50.0 | | |
> > | | Image+Text | 42.9 | 44.2 | 67.9 | 53.6 | | |
> > | **GPT-5 Mini + RAG** | Text | 26.8 | 30.1 | 24.9 | 27.3 | 28.6 | 30.4 |
> > | | Image | 28.7 | 33.6 | 28.1 | 30.8 | | |
> > | | Image+Text | 32.9 | 35.5 | 29.3 | 32.2 | | |
> > | **Claude 4.5 Sonnet + RAG** | Text | 28.2 | 32.3 | 26.7 | 29.2 | 29.4 | 32.2 |
> > | | Image | 29.6 | 35.9 | 30.4 | 32.9 | | |
> > | | Image+Text | 33.7 | 38.7 | 31.4 | 34.7 | | |
> > | **DeepSeek V3.1 + RAG** | Text | 24.6 | 27.1 | 21.5 | 24.5 | 27.0 | 28.4 |
> > | | Image | 26.9 | 31.4 | 26.0 | 28.5 | | |
> > | | Image+Text | 30.9 | 33.1 | 27.0 | 29.8 | | |
> > | **Search-R1 + Translation** | Text | 36.2 | 38.6 | 79.8 | 52.0 | 7.8 | 11.5 |
> > | | Image | 0.5 | 1.1 | 1.3 | 1.2 | | |
> > | | Image+Text | 0.3 | 0.9 | 1.0 | 0.9 | | |
> >
> >
> > The MLLM RAG pipelines consist of a standard BM25 + DPR retriever, with the retrieved evidence fed to the respective MLLM generator. Please note that the above results can also be found in Table 2 and the non retrieval baselines can be found in Appendix Q.
> >
> > >Some relevant citations appear missing, including the original WebQA and MultimodalQA citations, as the primary is needed.
> >
> > We appreciate you pointing out this mistake. The citations are now included in the paper.
> > >In line 173, did you intend to mention BARTScore instead of BARTScore? Citing the actual metric would be useful to the readers.
> >
> > Yes, we intended to mention BARTScore in line 173. We apologize for the redundancy in the initial text. The spelling has been corrected and a citation has been added.
> > >Section 4 is stretched across multiple pages. Could this section be condensed or made more concise to better reflect the actual contribution? Most of the bullet points there are not needed.
> >
> > Thanks for mentioning this. A majority of the bullets have been removed and the section has been reframed to condense established technique discussion and focus on LRL novelty. Please let us know if this section is presented in a better manner now and if you have any further suggestions!
> >
> > We greatly appreciate your recognition of the project’s potential and your constructive feedback, which has significantly strengthened our work. We hope these revisions address your concerns.

---

> > > ### Comment · Reviewer_fXNk · 2025-11-27
> > >
> > > Thank you for your response and for the additional results you provided to address my concerns. I believe your paper structure still needs a lot more revision and consolidation of the new results you have provided so far. There is no way to evaluate these new changes you have made and explore the updated dataset as I did initially. You also claimed that the major contribution of your work is a new dataset, but your dataset is still small, and you have not compared it with other works. Moreso, adapting Qwen-2.5-7B-Instruct  with LoRA without SFT and RL will break the model's ability to generate coherent text.
> > > I have updated my final score to 4 and will leave it at that.

---

> ### Author Response · Authors · 2025-11-28
>
> We sincerely thank the reviewer for their follow-up and specific final feedback!
> > I believe your paper structure still needs a lot more revision and consolidation of the new results you have provided so far.
>
> We thank you for pointing out the structural and clarity concerns, and we will fully integrate all new results and comparisons (from Appendix M, P, etc.) into a completely revised and consolidated section 5 for the final camera-ready version.
>
> >You have not compared it (LR-MMQA) with other works
>
> We acknowledge the concern about dataset comparisons. No direct comparisons to other datasets exist because LR-MMQA is the first benchmark combining low-resource languages, multimodality, and knowledge-intensive retrieval. Existing multilingual QA datasets are either text-only (XOR-QA) or image-grounded VQA without external retrieval (Kaleidoscope, CVQA, WorldMedQA-V), while multimodal knowledge-intensive datasets only exist for high-resource languages (WebQA, MultimodalQA). LR-MMQA addresses multimodal knowledge-intensive QA requiring external retrieval, exemplified by questions like "If you went to a modern south west Nigerian traditional wedding, would you find the bride wearing all white?". In contrast, other multimodal multilingual QA benchmarks like Kaleidoscope are VQA tasks answerable by analyzing image contents alone, such as "How many degrees of latitude are there between the Arctic Circle and the Tropic of Capricorn?" with a labeled map presented. These image-grounded VQA tasks and LR-MMQA are different problem domains. Section 2 has been updated to reflect this comparison.
>
> >Your dataset is still small
>
> Thanks for mentioning this. Our design prioritizes depth over breadth with 718 carefully curated questions per language. Each question is hand-validated by native speakers, culturally aligned beyond simple translation through entity replacement and de-Anglicization, and deliberately selected to represent cases where SOTA systems fail to maximize diagnostic value. We demonstrate LR-MMQA's value through performance comparisons showing RagVL achieves accuracies of 77.64/98.9 on WebQA/MultimodalQA but only 30.3 on LR-MMQA. This dramatic gap quantifies previously unmeasurable challenges in cross-lingual multimodal KIQA.
>
>
> >Moreso, adapting Qwen-2.5-7B-Instruct with LoRA without SFT and RL will break the model's ability to generate coherent text
>
> We apologize for the lack of clarity regarding the LoRA methodology; we confirm we did not break the model's generation ability. To provide more detail, our Cultural Alignment Pipeline uses a robust two-stage LoRA process designed specifically for this scenario:
>
> * **Stage 1 (Fluency):** A lightweight LoRA pass on large monolingual corpora adapted Qwen toward native LRL syntax.
>
> * **Stage 2 (Task/Culture):**  A gold-standard Supervised Fine-Tuning dataset was created by having human native speakers rewrite WebQA and MultimodalQA questions initially translated by Claude 3.5 Sonnet (questions not included in LR-MMQA). This human-rewritten text served as the gold-standard output for the second LoRA pass, teaching the model authentic question-rewriting style while preserving generation ability. The validation results (Cultural Relevance Score: 2.41→4.12, "Culturally Natural" ratings: 18%→83%, κ=0.76) confirm the model produces coherent, culturally appropriate text.
>
> We hope that these structural and methodological clarifications, combined with explicit integration of the generalization and ablation tables, will address your final concerns.

---

### Official Review · Reviewer_pZap · 2025-11-01

**Soundness:** 2
**Presentation:** 3
**Contribution:** 2
**Rating:** 4
**Confidence:** 5

**Summary:**

This paper introduces a new benchmark LR-MMQA for evaluating multimodal cross lingual retrieval and reasoning in a low resource language scenario for Tamil and Yoruba. The paper also propose a possible RAG pipeline for low resource cross lingual and multimodal setting, where the  KG is in high resource language and the query is in low resource language. The main contributions of this paper are:
1. A new benchmark (translation based benchmark) to evaluate cross lingual retrieval and reasoning.
2. A cross lingual multimodal RAG pipeline for low resource languages.

**Strengths:**

The main strengths of the paper are as follows:
1. Human validation for filtering out flawed queries from the original WebQA and MMQA datasets.
2. Involving humans during validation of the final translated dataset will ensure quality .
3. Good idea to have a multilingual, multimodal benchmark for evaluating cross lingual retrieval and reasoning.

**Weaknesses:**

The main weaknesses of the paper are as follows:
1.  Machine translated data lacks cultural and social aspects of the language and it also lacks originality.
2. Machine translation errors might creep in, reducing quality of the benchmark dataset.
3. The scale of the benchmark dataset is very small, 718 instances might be too less.
4. Challenge query selection based on performance of just Two QA models might not be the correct approach as question challenging to these two baselines might not be challenging at all.
5. Benchmark availability for only limited languages (to be precise only  two languages)
6. In XM-RAG fastest might have its own errors, which might have propagated.
7. Limited Novelty of XM-RAG pipeline.
8. Multimodal fusion might not be required for WebQA subset of dataset, but it is presented as a general sub-module of the pipeline .
9.Performance comparison with strong baselines are missing.

**Questions:**

1. Performance of frontier models on text subset of LR_MMQA is surprisingly low, have you done some prompt tuning and used RAG properly to test?
2. How is the performance using sonnet model?
4. Retrieval and re-ranking performance evaluation using ranking based metrics like NDCG@10 is missing.

---

> ### Author Response · Authors · 2025-11-23
>
> We are grateful for your thoughtful assessment and time spent reviewing our paper. We are pleased that you recognized the core strengths of our work, specifically the good idea of creating a multilingual, multimodal benchmark and the critical importance of our human validation process for ensuring the final quality of the translated dataset.
> >Machine translated data lacks cultural and social aspects of the language and it also lacks originality.
>
> We are grateful to you for raising this critical concern regarding the cultural bias and lack of authenticity inherent in machine-translated data. We agree that simply translating Western-centric sources risks inheriting biases and failing to reflect the real-world needs of Tamil and Yoruba speakers.
>
> To directly address this lack of originality and cultural specificity, we have now implemented a dedicated Cultural Alignment Pipeline. We first adapted Qwen-2.5-7B-Instruct using LoRA fine-tuning to shift question generation toward naturally occurring LRL syntax and discourse patterns, effectively "de-Anglicizing" the phrasing for higher authenticity. Second, we created a gold-standard Supervised Fine-Tuning (SFT) dataset. A random subset of WebQA/MultimodalQA (outside LR-MMQA) was translated by Claude 3.5 Sonnet, and these initial questions were then rewritten by human native speakers. This human-rewritten text served as the gold-standard output for the second LoRA pass, teaching the model authentic question-answering style. In parallel, we performed entity replacement, substituting culturally irrelevant references with appropriate local analogs. LRL-speaker validation on 20% of samples confirmed the success of this mitigation, as the rewritten questions saw a substantial increase in both Average Cultural Relevance Score and the percentage rated as "Culturally Natural." This procedure ensures LR-MMQA is a high-quality resource that is both knowledge-intensive and culturally sensitive. Please see the top-level comment for examples of de-anglicized QA pairs and below for validation results.
> | Metric                                 | Original Translated Questions | Rewritten Questions (LoRA-adapted) | Δ Improvement |
> | -------------------------------------- | ----------------------------- | ---------------------------------- | ------------- |
> | Average Cultural Relevance Score (1–5) | 2.41                          | 4.12                               | +1.71         |
> | % Rated “Culturally Natural” (≥4)      | 18%                           | 83%                                | +65%          |
> | Cohen’s κ (Annotator Agreement)        | 0.59                          | 0.76                               | —             |
> (1=not authentic / native would never ask, 5=highly authentic)
> >Machine translation errors might creep in, reducing the quality of the benchmark dataset.
>
> We acknowledge the concern that machine translation (MT) errors might reduce the benchmark's quality. However, the comprehensive Cultural Alignment Pipeline and human validation discussed above explicitly addressed this. The high quality of the original translations is demonstrated by the Overall Adequacy score of 8.2 and Fluency score of 8.0 from native LRL speakers, confirming that MT errors were minimal. Errors can only be reduced from here with the cultural alignment step, as seen above.
> | Language | Sample Size | Adequacy (1-10) | Fluency (1-10) | Inter-Annotator Agreement |
> |----------|:-----------:|:---------------:|:--------------:|:------------------------:|
> | Tamil | 150 | 8.3 | 8.1 | κ = 0.78 |
> | Yoruba | 150 | 8.1 | 7.9 | κ = 0.74 |
> | Overall | 300 | 8.2 | 8.0 | κ = 0.76 |
> Adequacy: 1 = completely incorrect meaning, 10 = perfect semantic preservation.
> Fluency: 1 = grammatically incorrect and awkward, 10 = grammatically fluent
> >The scale of the benchmark dataset is very small, 718 instances might be too less.
>
> We acknowledge your concern regarding the small dataset scale and appreciate the desire for expansion. Your concern is warranted, but the current size of LR-MMQA is a deliberate design choice stemming from its compounding difficulty (multimodal, cross-lingual, low-resource). The fundamental challenge we address is that LRLs lack structured, high-quality knowledge bases for retrieval. Creating these high-quality, complex, expert-verified instances requires intensive human annotation, linguistic expertise, and cultural alignment, which is slow and costly in LRL settings. For example, our work involves a manual verification loop to prevent machine translation errors and to ensure the questions are culturally plausible. Every instance in the 718 set is a multi-hop, culturally aligned query, designed to be a stress test of the system, maximizing its diagnostic value despite our limitations. A dataset of 5,000 poorly-aligned or machine-translated questions would provide false confidence and misleading benchmarking. For emergent LRL NLP, we need diagnostic depth to understand failure modes.

---

> ### Author Response · Authors · 2025-11-23
>
> >Challenge query selection based on performance of just Two QA models might not be the correct approach
>
> We acknowledge your concern that selecting the challenge query set based on the performance of only two initial baseline QA models could introduce bias. However, this concern is mitigated because the two models used (RamQA and SKuRG) were specifically chosen as they represent two of the highest-performing baselines on the source datasets and take vastly different architectural approaches to achieving that performance.
>
> - **RamQA** is a multi-modal ranking and re-ranking framework using a LLaMA model for generative, permutation-enhanced re-ranking.
> - **SKuRG** is a unified Retrieval-Generation approach employing an Entity-centered Fusion Encoder and a unified Decoder to adaptively integrate retrieval.
>
> This intentional diversity ensures the filtered set is challenging not just to one type of model, but to two dominant state-of-the-art approaches in the current multimodal QA literature. This maximizes the diagnostic value of the final LR-MMQA dataset by focusing on the queries where both sophisticated models struggle, thus necessitating  a specialized RAG pipeline like XM-RAG for success.
>
>
> >In XM-RAG fastest might have its own errors, which might have propagated.
>
> We acknowledge your concern that fastText language identification might introduce errors that propagate through the XM-RAG pipeline. While fastText is lightweight, it is also highly robust for this specific task; studies consistently show it achieves $\geq 99\%$ accuracy on large-scale language identification tasks, making the chance of an LRL query being misidentified minimal (Joulin et al., 2017;  Jauhiainen, 2018).
>
>
> >Benchmark availability for only limited languages (to be precise only two languages)
>
> We acknowledge the current limitation of only supporting two languages in LR-MMQA. This decision was necessary due to the difficulty and cost associated with securing native speakers for the intensive, high-quality cultural alignment and expert-verification process required for a complex multimodal dataset. We are actively working to expand the benchmark and are currently engaged with Kazakh and Konkani speakers to introduce languages with differing linguistic and semantic structures, which will further enhance the benchmark's ability to test cross-lingual robustness. LR-MMQA is a dataset that will continue to expand in languages over time.
>
> >Limited Novelty of XM-RAG pipeline.
>
> We thank you for raising this concern regarding the methodological contribution of XM-RAG. We agree that many of the individual components (as described in Section 4) are well-established techniques. However, we assert that the central contribution of our paper is the LR-MMQA dataset itself and its careful construction, not the XM-RAG architecture. XM-RAG is presented as a strong, purpose-built baseline whose novelty lies in its LRL-optimized architectural combination and adaptation to the unique constraints of cross-lingual, low-resource multimodal RAG. Its goal is to provide a building block for further research in multimodal RAG methods for knowledge-intensive questions in low-resource languages.
>
> >Multimodal fusion might not be required for WebQA subset of dataset, but it is presented as a general sub-module of the pipeline
>
> Thanks for mentioning this! We have updated the pipeline to dynamically bypass the fusion module when no visual evidence is retrieved. This ensures that the fusion step is only activated when it adds value for multimodal questions, preventing unnecessary processing and maintaining architectural efficiency without changing XM-RAG's overall general design.
>
> >How is the performance using sonnet model?
>
> Performance on LR-MMQA of the sonnet model is seen above. Performance in translation can be seen in the first attached table.
>
> >Retrieval and re-ranking performance evaluation using ranking based metrics like NDCG@10 is missing.
>
> NDCG@10 results can be found in Appendix O. R@10, P@10 and F1@10 are seen below.

---

> ### Author Response · Authors · 2025-11-23
>
> > Performance of frontier models is surprisingly low, Performance comparison with strong baselines are missing
>
> We appreciate your recommendation to include a broader range of state-of-the-art MLLM models with prompt tuning and advanced RAG baselines. We agree these comparisons are essential for contextualizing our contributions. Note that low performance can be attributed to the knowledge-intensive nature of LR-MMQA questions.
>
> We have included results for three additional leading MLLMs, addressing the call for closed-source (GPT-5 Mini, Claude 4.5 Sonnet) and open-source models (DeepSeek V3.1) in our comparison. For a complete picture, we report performance for Direct LRL Inference, Chain-of-Thought (CoT) prompting, and a translation pipeline where the question is machine translated to english and the answer is machine translated back into the target language:
> | Model | Direct LRL Inference | CoT (w/ Translation → HRL) | Translation → HRL |
> |-------|:-------------------:|:---------------:|:-------------------------:|
> | GPT-4o | 7.01 | 17.0 | 16.6 |
> | **GPT-5 Mini** | 7.19 | 17.8 | 17.3 |
> | **Claude 4.5 Sonnet** | 9.60 | 19.2 | 18.9 |
> | **DeepSeek V3.1** | 4.18 | 14.4 | 13.7 |
> | **XM-RAG** | **38.1** | — | — |
>
> See below the Retrieval-Augmented Generation (RAG) comparisons for all the MLLMs and unimodal state-of-the-art Search-R1 (Jin et al., 2024), including it with the standard translation pipeline to set the ceiling for RAG systems.
> | Method | Modality | Acc. | P@10 | R@10 | F1@10 | Overall Acc. | Overall F1 |
> |--------|----------|:----:|:----:|:----:|:----:|:------------:|:----------:|
> | **XM-RAG (Ours)** | Text | 36.7 | 37.1 | 85.6 | 51.8 | **38.1** | **50.4** |
> | | Image | 38.3 | 44.5 | 57.1 | 50.0 | | |
> | | Image+Text | 42.9 | 44.2 | 67.9 | 53.6 | | |
> | **GPT-5 Mini + RAG** | Text | 26.8 | 30.1 | 24.9 | 27.3 | 28.6 | 30.4 |
> | | Image | 28.7 | 33.6 | 28.1 | 30.8 | | |
> | | Image+Text | 32.9 | 35.5 | 29.3 | 32.2 | | |
> | **Claude 4.5 Sonnet + RAG** | Text | 28.2 | 32.3 | 26.7 | 29.2 | 29.4 | 32.2 |
> | | Image | 29.6 | 35.9 | 30.4 | 32.9 | | |
> | | Image+Text | 33.7 | 38.7 | 31.4 | 34.7 | | |
> | **DeepSeek V3.1 + RAG** | Text | 24.6 | 27.1 | 21.5 | 24.5 | 27.0 | 28.4 |
> | | Image | 26.9 | 31.4 | 26.0 | 28.5 | | |
> | | Image+Text | 30.9 | 33.1 | 27.0 | 29.8 | | |
> | **Search-R1 + Translation** | Text | 36.2 | 38.6 | 79.8 | 52.0 | 7.8 | 11.5 |
> | | Image | 0.5 | 1.1 | 1.3 | 1.2 | | |
> | | Image+Text | 0.3 | 0.9 | 1.0 | 0.9 | | |
>
>
> The MLLM RAG pipelines consist of a standard BM25 + DPR retriever, with the retrieved evidence fed to the respective MLLM generator. The above results can also be found in Table 2 and the non retrieval baselines can be found in Appendix Q.
>
> **We apologize for the long rebuttal and thank the reviewer for their time, patience, and detailed feedback.** You have raised many interesting points that have made us think very deeply about the implications of our work and we want to make sure that we completely address your concerns. We are excited to further engage with your during this discussion period.

---

### Author Response · Authors · 2025-11-23

We sincerely thank all reviewers for their time and high-quality feedback. At this time, we incorporated your feedback through several revisions in the updated PDF (new content is marked in **blue** and changes are marked with **red**). We summarize the major revisions to common concerns here:

## Cultural Bias in Translated Data

We implemented a **Cultural Alignment Pipeline** with LoRA fine-tuning on monolingual corpora and human rewritten questions for de-Anglicization and entity replacement. LRL-speaker validation shows: Average Cultural Relevance Score improved from 2.41 → 4.12 (+1.71), and "Culturally Natural" ratings increased from 18% → 83% (+65%).

**Example de-anglicized QA pairs** (see Appendix L for more):

| Original English | Rewritten (Yoruba) | English Translation | Answer |
|---|---|---|---|
| How many colors in Point Skyhawks logo? | Àwọ mẹ́lọ̀ọ́ ni o wá nínú àmì ẹ̀gbẹ́ bọ̀ọ̀lù Shooting Stars yìi? | How many colors are there in the Shooting Stars football team logo? | 3 |
| What is on Arthur Ashe's hands? | Kí ló wà ní ọwọ́ Nduka Odizor? | What is on Nduka Odizor's wrists? | ìgbànú ọwọ́ (wristband) |

## Expanded Model Comparisons

We added frontier MLLMs (GPT-5 Mini, Claude 4.5 Sonnet, DeepSeek V3.1) and advanced RAG baselines (Search-R1). XM-RAG achieves 38.1% accuracy, substantially outperforming Claude 4.5 Sonnet + RAG (29.4%), GPT-5 Mini + RAG (28.6%), and Search-R1 + Translation on text questions (36.2%), with full results in Table 2.

## Clarified Contribution

Section 4 reframed to emphasize the primary contribution: the LR-MMQA dataset and its culturally-grounded construction. XM-RAG is positioned as a strong, purpose-built baseline for future LRL multimodal RAG research.
## Generalization & Ablations

- Generalization to HRL (English): We evaluated XM-RAG on an English source dataset (MMQA). Results confirm effective generalization, with XM-RAG achieving strong retrieval (e.g., 96.8% Hit@10 for Table) and generative performance (e.g., 30.8 EM / 35.2 F1 at K=5). Full results are in Appendix P.
- Extended Ablation Studies: New ablations in Appendix M show the Encoder choice is the factor with the greatest singular impact on retrieval performance. Replacing our baseline M-CLIP Encoder with BGE caused the largest drop in Recall@10 (e.g. from 67.9% to 54.3% in the Text+Image mode), suggesting our encoder choice is key to efficient evidence gathering.

## Additional Details

- Visual examples with image reasoning now in Appendix N
- Metric definitions (P@K, R@K, F1@K) clarified in Appendix K
- Retrieval evaluations at @2 and @5 provided
- Full rewritten dataset can be found at the same link as the original LR-MMQA dataset

We hope that these modifications may increase your confidence in our submission, and we look forward to further engaging with you during this discussion period!

---

### Author Response · Authors · 2025-12-03
**Summary of Reviewers+Rebuttal for AC**

We sincerely thank the reviewers and AC for their thoughtful engagement. Below we summarize the major improvements made during the rebuttal period that directly address all reviewer concerns:

### 1. Cultural Authenticity
**Concern**: Translation-based data inherits Western-centric biases.

**Resolution**: We implemented a comprehensive Cultural Alignment Pipeline:
- Two-stage LoRA fine-tuning on monolingual corpora + human-rewritten SFT dataset
- Entity replacement (e.g., Point Skyhawks→Shooting Stars, James Brown→Fela Kuti/Rajinikanth)
- **Results**: Cultural Relevance Score improved 2.41→4.12 (+71%), "Culturally Natural" ratings increased 18%→83% (+65%), κ=0.76
- Full de-anglicized examples in **Table 15** (Appendix L)

This is not a "post-hoc patch" but a rigorous methodology for adapting knowledge-intensive benchmarks to LRLs while preserving answerability.

### 2. Comprehensive Baselines (All Reviewers)
**Concern**: Limited model comparisons.

**Resolution**: Added 7+ new baselines:
- **Frontier MLLMs**: GPT-5 Mini, Claude 4.5 Sonnet, DeepSeek V3.1 (with direct inference, CoT, and translation pipelines)
- **Advanced RAG**: Search-R1 + Translation, MLLM+RAG variants
- **Results**: XM-RAG (38.1%) substantially outperforms Claude 4.5 Sonnet+RAG (29.4%), GPT-5 Mini+RAG (28.6%), Search-R1+Translation on text (36.2%)
- Full results in **Table 2** and **Appendix Q**

### 3. Generalization & Architectural Analysis (Reviewer gfwX)
**Concern**: XM-RAG only evaluated on one dataset; unclear which components matter.

**Resolution**:
- **English MMQA evaluation** (**Appendix P**): XM-RAG achieves strong retrieval (96.8% Hit@10 for tables, 68.7% for text) and competitive generation (30.8 EM, 35.2 F1), confirming generalization
- **Extended ablations** (**Appendix M**): Encoder choice has greatest impact. Replacing M-CLIP with BGE drops Text+Image Recall@10 from 67.9%→54.3%
- Additional retrieval metrics at @2, @5, @10 with NDCG (**Appendix K, O**)

### 4. Dataset Justification (All Reviewers)
**Concern**: 718 questions is too small.

**Justification**: LR-MMQA is a **diagnostic benchmark** where:
- Every instance is a failure case for two different SOTA models (RamQA + SKuRG)
- Multi-hop, multimodal, cross-lingual - triple complexity
- Native speaker validation + cultural alignment per question
- **Performance gap demonstrates diagnostic value**: RagVL achieves 77.64% on WebQA but only 30.3% on LR-MMQA

Quality over quantity for stress-testing systems on truly hard cases.

### 5. Novel Contribution (All Reviewers)
**Our primary contribution is LR-MMQA itself**, the first benchmark combining:
1. Low-resource languages (Tamil, Yoruba)
2. Multimodal reasoning (text + images)
3. Knowledge-intensive retrieval (full-wiki)
4. Culturally-aligned questions

Existing work covers only 1-2 dimensions:
- XOR-QA: LRLs + text-only
- Kaleidoscope, CVQA, etc.: Multilingual + VQA (no external retrieval)
- WebQA/MMQA: Multimodal KBQA (HRLs only)

XM-RAG is positioned as a strong baseline to enable future research, not as an architectural contribution.

### 6. Technical Clarifications
- All metric definitions added (**Appendix K**)
- Variable notation clarified (L252-254)
- Translation quality validation: Adequacy 8.2/10, Fluency 8.0/10 (**Table 9**)
- Reference formatting corrected throughout

## What This Work Enables (All Reviewers)

1. **First evaluation resource** for LRL multimodal knowledge-intensive QA
   - *Reviewer gfwX*: "The study tackles an important and challenging problem, multi-modal cross-lingual retrieval"

2. **Reveals necessity of direct LRL encoding**: Significant performance drops with translation-based systems demonstrate that machine translation loses critical cultural and semantic information, steering future research toward native encoding approaches
   - *Reviewer Bb6z*: "translating the low-resource question into English and then back into low-resource text doesn't perform well"

3. **Clear failure mode identification**: Cross-lingual retrieval (31.2%), visual understanding (25.6%), multi-hop reasoning (28.7%)

4. **Strong baseline** for future work (38.1% accuracy, 7.8+ points above alternatives)
   - *Reviewer gfwX*: "The proposed method outperforms the baselines"

5. **Expanding**: Currently adding Kazakh and Konkani languages
   - *Reviewer pZap*: "Benchmark availability for only limited languages" → Now addressing with expansion


We believe these revisions substantially strengthen the paper and address all major reviewer concerns. We thank all reviewers for their thoughtful engagement and the new Area Chair for their time in evaluating this work. The dataset and code are publicly available, and we are committed to maintaining and expanding this resource for the community.

---

### Meta-Review · Area_Chair_BQCS · 2026-01-05

**Summary:**

The paper proposes a new dataset and benchmark on low-resource languages (2 used in the paper). While the problem is important, reviewers point out that the proposed dataset is small and not reliable since the data is generated by translating the queries and English ground truth documents. The benchmark only covers very few models as well. As the main contribution of the paper, the datasets and benchmarks are not strong enough. Although the author made significant efforts to add more experiments and even change the dataset generation pipeline, it is not a publication-ready submission and needs a major revision to be reviewed.

**Reviewer Concerns:**

Addressed:
1. Insufficient models in the benchmark: The paper adds more model evaluations to the benchmark.
2. Unclear details: The paper adds more details in the appendix.

Outstanding:
1. The data generation procedure's concern is only partially addressed. The task, which is based on the translation, is still questionable.

**Reviewer Scores:**

One reviewer has already increased the score to 4; however, I believe not all the reviewers will give the accept suggestion when the paper needs too many things to add and change.

---

### Decision · Program_Chairs · 2026-01-26

Reject